

# Surge dynamics and lake outbursts of Kyagar Glacier, Karakoram

Vanessa Round[1], Silvan Leinss[2], Matthias Huss[1,3], Christoph Haemmig[4], and Irena Hajnsek[2]

[1]Laboratory of Hydraulics, Hydrology and Glaciology (VAW), ETH Zurich, 8093 Zurich, Switzerland
[2]Institute of Environmental Engineering, ETH Zurich, 8093 Zurich, Switzerland
[3]Department of Geosciences, University of Fribourg, 1700 Fribourg, Switzerland
[4]GEOTEST AG, Zollikofen, Switzerland

*Correspondence to:* V. Round (vround@hotmail.com) / S. Leinss (leinss@ifu.baug.ethz.ch)

**Abstract.**

The recent surge cycle of Kyagar Glacier, in the Chinese Karakoram, caused formation of an ice-dammed lake and subsequent glacial lake outburst floods (GLOFs) exceeding 50 and 40 million $m^3$ in 2015 and 2016, respectively. GLOFs from Kyagar Glacier reached double this size in 2002 and earlier, but the role of glacier surging in GLOF formation was previously unrecognised. We present an integrative analysis of the glacier surge dynamics from 2011 to 2016, assessing surge mechanisms and evaluating the surge cycle impact on GLOFs. Over 80 glacier surface velocity fields were created from TanDEM-X, Sentinel-1A and Landsat satellite data. Changes in ice thickness distribution were revealed by a time series of TanDEM-X DEMs. The analysis shows that during a quiescence phase lasting at least 14 years, ice mass built up in a reservoir area at the top of the glacier tongue and the terminus thinned by up to 100 m, but in the two years preceding the surge this pattern reversed. The surge clearly initiated with the onset of the 2014 melt season, and in the following 15 months velocity evolved in a manner consistent with a hydrologically-controlled surge mechanism with dramatic accelerations coinciding with melt seasons, winter deceleration accompanied by subglacial drainage, and rapid surge termination following the 2015 GLOF. Rapid basal motion during surging is seemingly controlled by high water pressure caused by input of surface water into either an inefficient subglacial drainage system or unstable subglacial till. Over 60 m of thickening at the terminus caused potential lake volume to increase more than 40-fold since surge onset, to currently more than 70 million $m^3$, indicating that lake formation should be carefully monitored to anticipate large GLOFs in the near future.

## 1 Introduction

Glacier surges are dynamic instabilities affecting about 1% of glaciers worldwide (Jiskoot et al., 2000). They consist of periodically alternating long quiescent phases, characterised by years to decades of slow flow, and short active surge phases, characterised by months to years of acceleration and mass transport down the glacier (Meier and Post, 1969). During the active surge phase the glacier typically experiences dramatic lengthening or thickening at the terminus with potentially hazardous consequences, in particular ice-dammed lake formation (Harrison et al., 2014).

While surging glaciers in North America and Svalbard have been investigated in considerable detail, the large concentration of surge-type glaciers existing in the central Asian mountains, including the Karakoram (Copland et al., 2011), have been



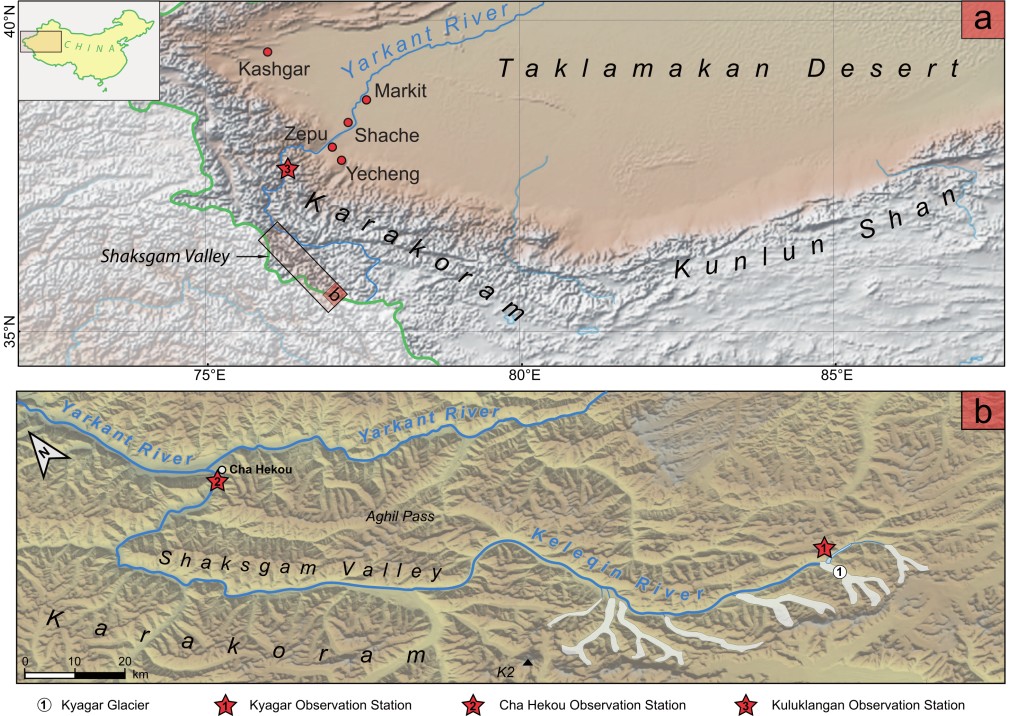

**Figure 1.** Location of (a) the Shaksgam Valley on the north side of the Karakoram Mountains in Western China and (b) Kyagar Glacier in the Upper Shaksgam Valley. Observation stations at Kyagar Glacier, Cha Hekou and Kuluklangon are indicated. The main flood impacts occur after the Yarkant River leaves the mountains near the Kuluklangon station.

relatively little studied. The recent unprecedented collapse of two surging glaciers in Tibet (GAPHAZ, 2016) highlighted the rather unknown nature of glacier surging in High Mountain Asia. Improved understanding of surge glacier dynamics in this region is crucial to anticipating glacier behaviour and hazard development in the future.

Kyagar (Keyajir) Glacier, situated on the northern slopes of the Karakoram Mountains, occasionally causes glacial lake

5  outburst floods (GLOFs) with devastating impacts on downstream communities along the Yarkant River in northwestern China (Zhang, 1992; Hewitt and Liu, 2010; Haemmig et al., 2014). The lake forms when ice at the glacier terminus impounds the river in the Upper Shaksgam Valley. Owing to the remote location of Kyagar Glacier, about 450 km upstream of the Yarkant floodplain (Fig. 1), the origin of these floods was poorly understood in the past and they arrived without warning (Zhang, 1992; Hewitt and Liu, 2010). An automated monitoring station was placed at Kyagar Glacier in 2012 to assess lake

10  formation (Haemmig et al., 2014), at which time there was no lake because the river flowed through subglacial channels at the terminus. From mid-2014, camera images from the station showed dramatic thickening of the glacier terminus, followed by lake formation. This rapid thickening indicated a possible glacier surge. While it was already recognised in the 1990s that Kyagar Glacier sometimes dammed the river in the Upper Shaksgam Valley and that there had been periods of advance or





thickening in the late 1920s and 1970s (Zhang, 1992) and 1990s (Hewitt and Liu, 2010), the possibility of Kyagar being a surge-type glacier wasn't realised until recently (Gardelle et al., 2013; Haemmig et al., 2014) and no surge of the glacier has ever been documented.

Hazardous lake formation at Kyagar Glacier is closely linked to the glacier's surge behaviour, as increased ice thickness

and deformation may push closed subglacial channels at the terminus through which the river normally flows and a higher ice dam at the terminus enlarges potential lake size. Such lakes usually fill during the summer months and empty through rapid subglacial drainage in late summer or autumn (Hewitt and Liu, 2010; Chen et al., 2010). An investigation of historic GLOF occurrences from Kyagar Glacier shows that there have been three main periods of flooding in the last 60 years, with peak volumes larger than 130 million m$^3$ in 1961, 1978 and 1999 (Fig. 2). These periods of increased GLOF activity may coincide

with glacier surges and indeed there was suspected thickening between 1987 and 1999 (Hewitt and Liu, 2010) and in the late 1970s (Zhang, 1992).

Surging affects both temperate and polythermal glaciers with a variety of geometries and settings (Clarke et al., 1986; Clarke, 1991; Jiskoot et al., 2000) and on vastly different timescales. The general mechanism is as follows: an unstable profile develops during quiescence, as mass accumulation higher on the glacier and mass loss over the lower part of the glacier cause

steepening of the surface and increase of basal shear stresses to a point at which a surge occurs through dramatically accelerated basal sliding (Raymond, 1987). The proposed mechanisms by which the accelerated basal sliding occurs are various and not completely understood, particularly because the subglacial environment is so difficult to observe. A switch in basal thermal conditions has been identified as a surge mechanism for some polythermal glaciers, with surging occurring when cold basal conditions switch to temperate (Clarke et al., 1984; Murray et al., 2000; Fowler et al., 2001). On the other hand, for temperate

glaciers and many polythermal glaciers that are already temperate at the base (Sevestre et al., 2015), surging has been explained by a hydrological switch mechanism, by which a surge occurs when the subglacial drainage system becomes inefficient, raising subglacial water pressure and facilitating rapid sliding (Kamb et al., 1985; Björnsson, 1998). Rapid deformation within subglacial till, in response to disturbance of the hydrological system within the till and increased effective water pressure, has also been proposed as an important possible surge mechanism, and is the largest uncertainty in surge understanding (Boulton

and Jones, 1979; Truffer et al., 2000; Harrison and Post, 2003). In all cases, a number of positive feedback mechanisms may enhance basal motion during a surge, for instance feedbacks between deformation, frictional heating and subglacial water pressure (Weertman, 1969; Clarke et al., 1984; Sevestre et al., 2015).

Glacier surging in the Karakoram region has mainly been studied by satellite remote sensing, owing to the difficulty of field access, in particular to identify surge glaciers through visible morphological features (Barrand and Murray, 2006; Copland

et al., 2011) and to observe surge dynamics through surface velocities (Quincey et al., 2011, 2015; Mayer et al., 2011). Quincey et al. (2011) interpreted a lack of seasonal control on surge initiation as an indication of thermally controlled surges, whereas Mayer et al. (2011) proposed a hydrological switch mechanism for Gasherbrun Glacier. Quincey et al. (2015) concluded that Karakoram glacier surging must be quite heterogeneous with a spectrum of surge mechanisms at play, having observed surges exhibiting a surge-front like down-glacier acceleration as well as surges showing simultaneous glacier-wide acceleration. In

the nearby West Kunlun Shan, two glacier surges showed a clear seasonal modulation of velocities during the active phase with




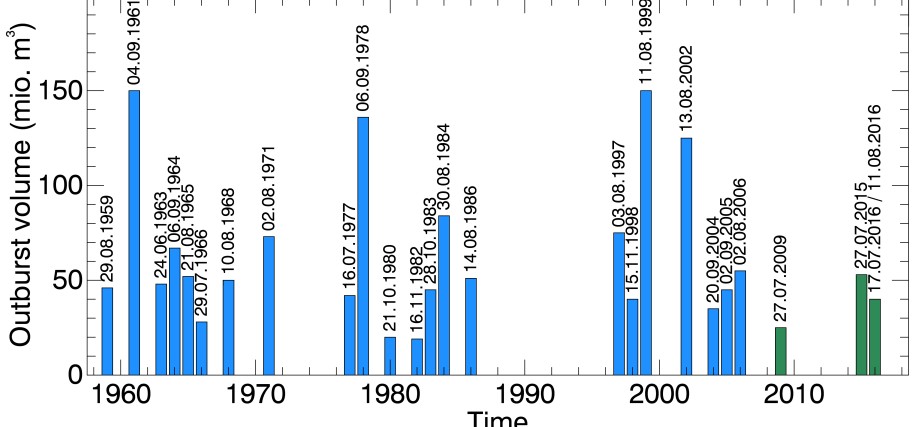

**Figure 2.** Historical GLOF volumes from Kyagar Glacier since the 1960s. Volumes from 1959 – 2006 are redrawn after Zhang (1992) and Chen et al. (2010). Volumes from 2006 – 2016 are estimated from lake extent on satellite images.

winter velocities up to 200% higher (Yasuda and Furuya, 2015). The main limitations of these studies were data gaps meaning that various stages of the surge development, such as surge initiation, weren't observed, and changes to ice mass distribution during surging also weren't investigated. For such an investigation, digital elevation models (DEMs) from before and after the surge would be required.

In this study the combination of optical and synthetic aperture radar (SAR) satellite data reveals the lead up, the onset and termination of the surge, as well as velocity modulations in relation to the seasonal cycle during the surge phase. A DEM time series exposes the ice mass distribution changes caused by the surge and allows us to examine the mass build-up which ultimately drives the surge. Our analysis of the most up-to-date available satellite tools provides a synthesis of the dynamics of a Karakoram glacier in unprecedented detail, showing the relationships between surging and external factors such as seasonal
melt cycles and lake drainage events.

In addition, we assess the impact of surging on the GLOF hazard posed by Kyagar Glacier in the recent past and into the future. GLOF hazard is largely determined by the lake volume and its drainage rate (Björnsson, 2010). The presented time series of glacier DEMs allows for the estimation of the potential lake volume and projection of potential GLOF volumes in the near future, and high-resolution satellite images reveal the drainage mechanism.

## 2  Study site and data

### 2.1  Study site

Kyagar Glacier is a polythermal glacier spanning from 4800 to over 7000 m a.s.l., consisting of three upper glacier tributaries 6–10 km in length which converge to form an 8 km long glacier tongue, approximately 1.5 km wide (Fig. 3, Haemmig et al., 2014). The total glacier area is 94 km$^2$ (Randolph Glacier Inventory Version 5.0, 2015) and the average surface slope is ap-





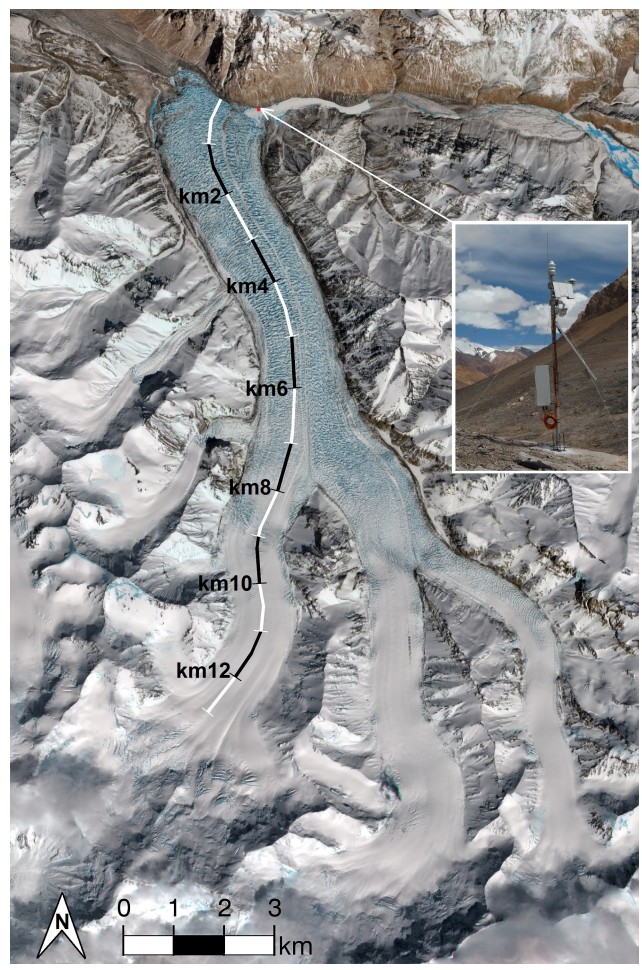

**Figure 3.** Optical image of Kyagar Glacier on 29.03.2016 from the ESA Sentinel-2A satellite. The glacier-dammed lake of approximately 5 million m³ is visible to the east of the glacier terminus. The curved scale bar up the west branch indicates the longitudinal profile used for surface velocity and elevation analysis, and the inset shows the monitoring station located about 500 m upstream of the glacier terminus.

proximately 2° over the tongue and 4.5°–20° over the branches above the confluence. The surface of the glacier tongue is characterised by ice pinnacles (Fig. 4) up to 40 m high and as narrow as 10 m, indicating cold ice and low shear deformation (Haemmig et al., 2014).

The tongue of Kyagar Glacier is most likely carved into the brown/black shales and cherty limestones of the 3 km thick Perminan-Jurassic Shaksgam sedimentary formation, while the mountain range forming the southern margin of the glacier catchment consists of the Aghil formation limestone and perhaps dolomite (Desio et al., 1991). The Shaksgam Valley follows the Shaksgam fault which passes under the terminus of Kyagar Glacier (Searle and Phillips, 2007).

Fieldwork at Kyagar Glacier is limited because of its remoteness and political restriction of access. Since a Sino–Swiss expedition in 2012 (Haemmig et al., 2014), in situ observations became available from an automated monitoring station 600 m





upstream of the Kyagar Glacier terminus (Fig. 3), which operated from 7 September 2012 until being drowned by the growing lake on 29 June 2015. According to this monitoring station, air temperatures typically range between 0 and 10°C in summer and $-15$ and $-5$°C in winter. The northern Karakoram is largely influenced by westerly weather patterns and snow accumulation mainly in winter, while rainfall (at lower altitudes) peaks between May and September (Kapnick et al., 2014). Balanced or

slightly positive mass balances for Karakorum glaciers between 1999–2011 (Gardelle et al., 2013) contradict global trends of decreasing glacier mass balance in line with global warming, but may be explained by regional increases in winter precipitation (Kapnick et al., 2014).

## 2.2 Data

In situ data from the automated observation station included daily camera images of the glacier terminus, showing the upstream

face of the ice dam (Fig. 4) as well as a wide 'fish eye' view of the lake basin. Meteorological variables included air temperature and precipitation amount and type, among others, recorded at hourly intervals until the station became submerged on 29 June 2015. Further meteorological data and river water level measurements were available from monitoring stations on the Yarkant River located at Cha Hekou and Kuluklangon, 190 and 500 km downstream from Kyagar Glacier, respectively (Fig. 1).

Three different satellite systems were used to determine surface velocities from the end of 2011 to mid-2016: the SAR

systems Sentinel-1A (S1A) and TanDEM-X and the Landsat-8 optical system. TanDEM-X is a formation of two tandem satellites, TanDEM-X and TerraSAR-X, data from both of which are used for all velocity and elevation analyses. In addition, Sentinel-2 images were used for visual assessment of lake formation but not for velocity analysis. Acquisition details of the three main satellite systems are presented in Table 1 (for a complete list of acquisitions see supplementary material).

## 3 Methods

### 3.1 Image co-registration

All satellite scenes were co-registered to a common master scene to allow for accurate comparison of images from the same orbit. The master scene for S1A and Landsat-8 images was the first available image from each orbit, while for TanDEM-X the master was updated progressively as an average of all previously co-registered slave scenes. The scenes used for image co-registration covered an area of approximately $30 \times 50$ km$^2$ extending north from Kyagar Glacier. Local offsets with sub-pixel

accuracy were calculated for patches of $512 \times 512$ pixels$^2$. To remove offsets resulting from patches covering moving glaciers, a planar function was fitted to the offset-fields and large outliers were removed before again fitting a planar function to the filtered offset fields. The co-registered slave scenes were then resampled according to the fitted function, resulting in a stack of images with sub-pixel co-registration accuracy.




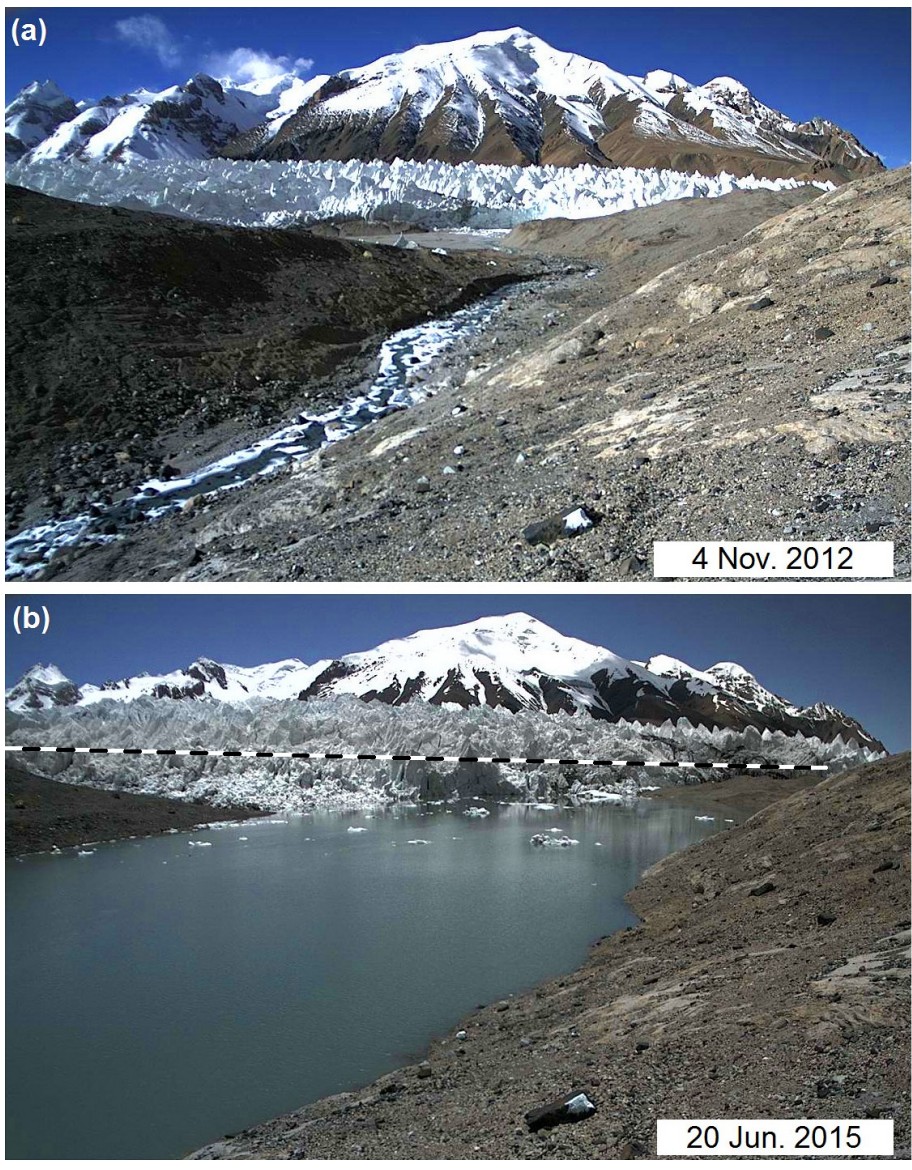

**Figure 4.** Images from the observation station upstream of Kyagar Glacier's terminus from (a) before and (b) during the surge. The glacier, flowing from left to right, blocks the flow of the river and causes lake formation during the surge. The dashed line in (b) indicates the ice dam height from 2012 (a), highlighting the dramatic thickening at the terminus. *Images: GEOPRAEVENT AG.*





**Table 1.** Summary of data products acquired by the three listed satellite systems used for this study

|  | **TanDEM-X** | **Sentinel-1A** | **Landsat-8** |
|---|---|---|---|
| **Authority** | DLR | ESA | USGS |
| **Data access** | proposal XTI_GLAC6780 | Open online | Open online |
| **First data available** | January 2008 | October 2014 | 2013 (1972 older versions) |
| **Spectral band (wavelength $\lambda$)** | X-band (3.1 cm) | C-band (5.4 cm) | visible - IR (0.43 – 12.51 $\mu$m) |
| **Processed data product** | CoSSC (Level 1b) | GRDH IW (Level 1) | Panchromatic (B8) and NIR (B5) |
| **Sampling resolution (m$^2$)** | $2.02 \times 2.18$ [1] ($75D$[2]) $2.17 \times 2.21$[1] ($98A$[2]) | $10 \times 10$ [3] | $15 \times 15$ |
| **Orbit height** | 514.8 km | 693 km | 705 km |
| **Incidence angle $\theta$** | 42.2–43.5° (75D) 38.2–39.5° (98A) | 32.1-32.3° | 90° |
| **Acquisition time (UTC) [5]** | 00:54 (75D) 12:46 (98A) | 00:57 (descending) 00:49 (descending) | 05:29[4] |
| **Orbit revisit** | 11 days | 12 days | 16 days |

1) single look complex (SLC) single-look sampling resolution (range × azimuth).

2) 75D: orbit 75 descending (flying north to south), 98A: orbit 98 ascending (flying south to north); both having the view direction to the right.

3) Ground Range Detected (GRD) multi-looked resolution (range × azimuth).

4) Orbit: Path 148, Row 35.

5) Local daytime at Kyagar Glacier on 21. June 2016: sunrise: 23:36 UTC; sunset: 14:10 UTC.

**Table 2.** Patch size and patch spacing used for velocity determination.

|  | Patch size | | Patch spacing | |
|---|---|---|---|---|
|  | (pixels) | (meters$^2$) | (pixels$^2$) | (meters) |
| TanDEM-X | $256 \times 256$ | $512 \times 512$ | 32 | 70 |
| Sentinel-1A | $64 \times 64$ | $640 \times 640$ | 18 | 180 |
| Landsat | $32 \times 32$ | $480 \times 480$ | 12 | 180 |

## 3.2 Glacier surface velocity

Glacier surface velocities were determined using offset tracking, through which the ground offset between patches of co-registered repeat-pass satellite image pairs is computed (Strozzi et al., 2002; Luckman et al., 2007). Patch-wise intensity cross-correlation was applied to pairs of SAR images. For Landsat optical data, phase cross-correlation was used to better

5  deal with variable illumination conditions (Zitova and Flusser, 2003). The resulting offset field covering the glacier and its surroundings was then converted to surface velocity by dividing by the elapsed time between the paired images and scaling by ground range resolution. Longitudinal velocity profiles were determined along a manually determined central glacier flowline (as shown in Fig. 3) in the velocity offset patch coordinates.





The patch size and spacing are presented in Table 2. Patch sizes were selected to optimise the superior correlation ability of larger patches with the superior spatial resolution of smaller patches. Larger patches were required for the SAR systems, despite their finer resolution, to compensate for radar speckle. Velocity fields were filtered to remove offsets calculated with low correlation quality, as determined by the height of the correlation-function peak over the noise. Offsets with high divergence

from neighboring values and outliers with velocities 50% larger than the maximum offset over the glacier were also removed.

The accuracy of the offset tracking procedure was assessed by calculating the patch offsets over a $1 \times 2 \, \text{km}^2$ area of stable ground next to the glacier terminus. Since no offsets are expected over stable ground, offsets represent local inaccuracies in the co-registration of images caused by slight changes in imaging geometry and, hence, scene projection, as well as the inherent inaccuracy in the sub-pixel determination of the correlation-function peak. The root-mean-square error for the offsets

over stable ground was 0.08 pixels or less for almost all image pairs from the three satellite systems, similar to the 0.05 pixel error estimated by Strozzi et al. (2002). Larger errors were experienced for some Sentinel-1A scene pairs, in which there were slight changes in imaging geometry and in particular erroneous east-west offsets resulting from warping by the Ground Range Detected projection to a terrain height varying strip-wise in the azimuth direction (Bourbigot et al., 2016, p. 12).

The glacier surface was assumed horizontal when converting pixel offsets to ground velocities. The subsequent velocity

error from this assumption was only about 0.06% over the 2° sloping glacier tongue and 0.4% over the 5° slope just above the confluence. Due to the side-looking radar imaging geometry, steep slopes in the range direction show distorted velocities. However as the glacier is not very steep and flows predominantly in the azimuth direction, this is not a problem.

### 3.3 Digital elevation models

Digital elevation models were derived using data from the TanDEM-X satellite formation (Krieger et al., 2007, 2013) using

single-pass SAR interferometry. SAR interferometry allows accurate DEM generation if the absolute interferometric phase can be successfully determined from the wrapped interferometric phase measured between 0 and $2\pi$. Determination of the absolute phase requires phase unwrapping algorithms (e.g. Goldstein et al., 1988; Zebker and Yanping, 1998). Phase unwrapping can be simplified by subtracting a synthetic interferogram, based on a reference DEM, before phase unwrapping and adding the synthetic interferogram back to the unwrapped interferogram (e.g. Dehecq et al., 2015). Subtraction of the reference DEM

also helps minimising phase-wrapping errors which can easily be recognised in the interferograms when an accurate reference DEM is used. If the phase difference to the measured data does not exceed $2\pi$, phase unwrapping can even be avoided entirely.

The phase gradient and hence the DEM accuracy depends on the perpendicular interferometric baseline $B_\perp$, which is the across-track separation between both SAR sensors (here the satellites TerraSAR-X and TanDEM-X) perpendicular to the line-of-sight. Large baselines provide a better height accuracy with phase cycles of $2\pi$ corresponding to smaller height of ambiguity

(HoA, see p. 3320 and Eq. 37 in Krieger et al., 2007), but on the other hand large baselines are more prone to phase unwrapping errors and signal decorrelation due to scattering volumes (Zebker and Villasenor, 1992) and also due to noise contained in the reference DEM.

The DEMs used for this study were generated with the help of a reference DEM, based on the 30 m resolution Shuttle Radar Topography Mission (SRTM) DEM (global version 3.0, 2015) which was updated and corrected by the average of eight





TanDEM-X DEMs from orbit 75D between 12 Oct. 2015–28 Dec. 2015. Phase unwrapping errors could be avoided due to the very short baselines of 19–29 meters giving large HoAs of 250–400 m. The interferograms were filtered using the adaptive interferogram filter proposed by Goldstein and Werner (1998). After phase unwrapping and conversion to height, the height corrections were averaged and added to the SRTM DEM to form the reference DEM which was downsampled to a resolution

of $8 \times 8 \, \mathrm{m}^2$. DEMs for each acquisition from orbit 75D were created by converting the phase difference against the reference DEM into a height change $\Delta h$, which was then added to the reference to obtain an absolute DEM for each acquisition date.

DEMs over the tongue of the glacier could only be calculated with baselines $B_\perp < 200$ m because the pinnacled surface structure of Kyagar Glacier caused such a strong decorrelation in the SAR interferograms that no reliable phase values could be extracted. The strong decorrelation was caused by the extremely rough glacier surface topography with ice pinnacles up to

40 m high and 20–40 m apart (estimated from shadow lengths and the observations from Haemmig et al. (2014)), which cause frequent phase wraps within the coherence window of $15 \times 15 \, \mathrm{m}^2$ for HoAs below 20 m ($B_\perp > 200$ m).

The generated DEMs contain errors from processing uncertainties as well as from microwave penetration into snow. Processing uncertainties include phase noise due to low correlation in the interferograms, global offsets due to geometric errors, and errors of the SRTM DEM. Errors due to phase noise were estimated from the standard deviation between the eight DEMs

used to form the reference DEM. The standard deviation was below 4 m for 90% of pixels with coherence values $> 0.3$ (mean standard deviation 2.5 m) and global offsets were below $\pm 1$ m. The SRTM DEM is specified with an absolute vertical accuracy of about 10 m (Farr et al., 2007), but for comparison of DEMs systematic vertical shifts or tilts were corrected for by referencing DEMs to a common reference height such that the height difference in the flat valley bottom near the tongue of Kyagar Glacier was zero. The remaining relative error between different DEMs was estimated from four flat valley planes and resulted

in a maximum height error of $\pm 1$ m (standard deviation 0.65 m).

The error due to microwave penetration into dry snow can reach up to 6 m (Dehecq et al., 2015) for a microwave frequency of 9.65 GHz (TanDEM-X), but penetration is negligible over wet snow (more than 1% volumetric water content, Leinss et al., 2015, Fig. 5). Microwave penetration leads to potential underestimation of the actual surface height over dry snow and ice surfaces. For Kyagar Glacier, penetration depths of up to two meters have been estimated as follows: firstly, the radar

backscatter signal was analysed to distinguish between dry and wet snow conditions (Nagler and Rott, 1998; Small, 2012; Nagler et al., 2016), and then the apparent height difference between images with wet and dry conditions was identified (Fig. 5).

Over the tongue of Kyagar Glacier the backscatter intensity, and hence penetration, changed very little between seasons ($<5$ dB). This can be explained by the surface of the tongue being snow-free ice the majority of the time with fresh snowfall a

rare event, and because the surface roughness dominates the backscatter signal. Over the accumulation basin, large seasonal changes in backscatter intensity indicate changing water content and thus penetration. Backscatter decreased by more than 10 dB between images 22 days apart at the onset of snowmelt in 2015, and a height difference of less than 2 m was calculated from two large baseline interferograms from before snowmelt and at the onset of snowmelt. A similar 2 m height difference was observed using small baseline DEMs from Aug. and Sep. 2015 (Fig. 5a, wet snow, low backscatter) compared to December

2015 (Fig. 5b, dry snow, high backscatter). These observations indicate a TanDEM-X penetration depth of 2 m or less in dry




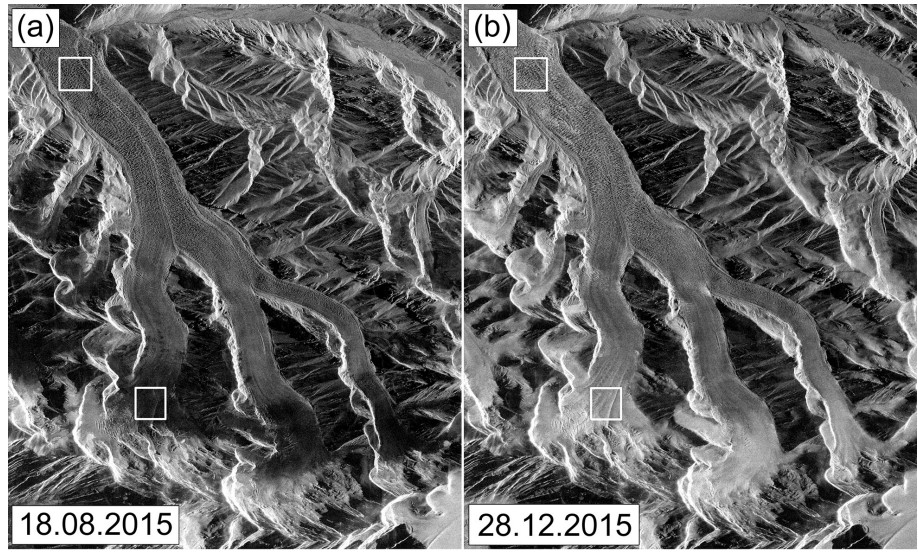

**Figure 5.** TanDEM-X radar backscatter images highlighting differences in backscatter intensity in (a) summer versus (b) winter. In the upper accumulation basin (white square at bottom of image) the dark areas in August 2015 (a) indicates low backscatter intensity from wet snow, while the high backscatter intensity in winter 2015 (b) indicates dry snow. A 2 m height difference in the area indicates approximately 2 m penetration into the dry snow. Over the glacier tongue (white square at top of image), little change in backscatter intensity indicates bare ice and, hence, very small penetration depth differences. Image data provided by DLR.

snow conditions over the upper glacier and negligible penetration over the glacier tongue. The relatively small penetration depths are likely a result of thick firn layers and ice inclusions in the accumulation area, as also observed by Dehecq et al. (2015).

The penetration error in the SRTM DEM should be slightly larger than for the TanDEM-X DEMs, as the SRTM DEM

was acquired with a C-band radar with 5.3 GHz (Farr et al., 2007) during winter (Feb. 2000). For C-band radars, expected penetration depths are 1–2 m into exposed ice (Rignot et al., 2001) and 5–10 m into dry snow (Rignot et al., 2001; Fischer et al., 2016). However, because in the accumulation area the penetration for X-band is <2 m, we estimate a penetration of <4 m in C-band (cf. Fig. 9 in Fischer et al., 2016), and over the glacier tongue 1-2 m.

In summary, systematic shifts are removed when comparing DEMs but differences in penetration must be considered in

particular when comparing the SRTM to the TanDEM-X DEMs or when comparing DEMs from different seasons. Over the glacier, tongue penetration errors are < 2 meters and over the accumulation area they are estimated to be <4 meters.

### 3.4   Calculation of positive degree days

Positive degree days (PDD) were calculated as a proxy for potential melting, through summing the magnitude of positive air temperatures, weighted by the fraction of a day which they represent (Vaughan, 2006) such that one PDD would be one day at

a temperature of 1°C. The hourly air temperature data from the station at Kyagar Glacier were used for computing PDDs in



2013 and 2014. Data from the downstream Cha Hekou observation station (Fig. 1) were used to estimate PDDs at the Kyagar Glacier in 2015 and 2016, to replace data from the Kyagar observation station which had been submerged. PDDs from the downstream station were scaled by using average monthly PDD offsets to the Kyagar Glacier station data from 2013 and 2014.

### 3.5 Lake volume estimation

Lake volumes were calculated using the DEM of the empty lake basin from TanDEM-X data acquired on 18 Aug. 2016, together with the lake extent and thus lake surface altitude from optical (Landsat or Sentinel-2) or SAR backscatter images (Sentinel-1A and TanDEM-X). In addition, the initial lake formation during the winter of 2014/15 was observed by the in situ camera as the small initial volumes were not seen on the satellite images but were important for assessing possible subglacial drainage. Potential lake volumes, and hence flood potential, were estimated by calculating the lake volume which would result if the lake basin would be filled to the 90% of the ice dam height, as determined by the DEMs of the glacier terminus.

## 4 Results

### 4.1 Glacier surface velocities

More than 80 surface velocity fields over Kyagar Glacier from September 2011 to August 2016 capture 2.5 years building up to the surge, the initiation of the surge in May 2014 and several periods of acceleration and deceleration in the two years following the main surge phase. Pre-surge velocity is represented in Fig. 6, while Figs. 7 and 8 are maps of surface velocity during the surge onset and main development, and Fig. 9 depicts the temporal and spatial velocity profiles over the entire study period in a 2D colour diagram. A complete set of surface velocity maps from all three satellite systems are provided in the supplementary material.

In the 2.5 years before surge onset, a gradual but clear acceleration occurred, greatest over the middle of the glacier tongue (between km 3 and km 6) with an increase in velocity from $0.1 \, \mathrm{m \, d^{-1}}$ in winter 2011/12 to over $0.4 \, \mathrm{m \, d^{-1}}$ in winter 2013/14 (Fig. 6). The location of the maximum velocity moved from above the confluence at km 10 at the end of 2011 to over the glacier tongue at km 5 in 2013/2014. Aside from this shift, the acceleration occurred quite uniformly over the whole glacier tongue and did not show an obvious acceleration front as observed for some other Karakoram Glaciers (Mayer et al., 2011; Quincey et al., 2015). The presence of seasonal modulation could not be assessed due to the coarse temporal resolution of the six pre-surge acquisitions, but it can be seen that acceleration continued over the winter immediately before surge initiation (Fig. 6, fastest velocity profile, from Oct. 2013–Feb. 2014). This gradual pre-surge acceleration may indeed have already been under way prior to 2011, with acceleration between annual velocities from 2004/2005 to 2010/2011 based on Landsat velocity analysis by Heid and Kääb (2012).

The pre-surge acceleration appears insignificant in comparison to the main surge phase, which started at the end of April 2014. Rapid acceleration first became evident between 28 Apr. and 30 May 2014 (Fig. 7a–b), with a doubling of maximum velocity from $0.5 \, \mathrm{m \, d^{-1}}$ to over $1 \, \mathrm{m \, d^{-1}}$ within a 32-day period. Velocities continued increasing steadily (Fig. 7c) to a peak of



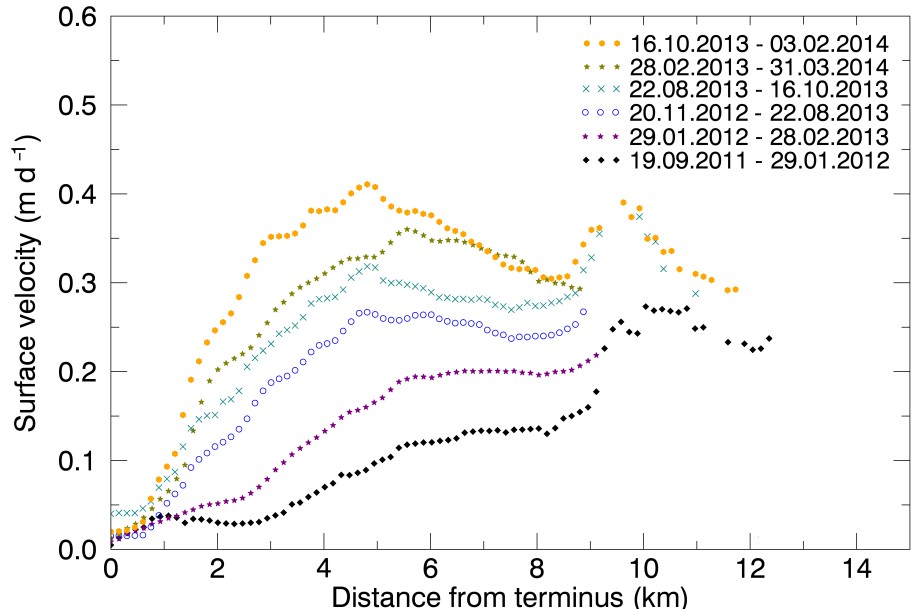

**Figure 6.** Longitudinal surface velocity profiles showing (from bottom to top) the pre-surge acceleration that occurred in the 2.5 years before the main surge onset. The profiles, derived from TanDEM-X data, follow the longitudinal path from Fig. 3. Gaps above km 9 indicate failed velocity calculation owing to the poor surface contrast providing no clear correlation. The labels state the time period over which each velocity calculation was averaged, in this case ranging from 2–13 months.

almost $2.5\,\mathrm{m\,d^{-1}}$ (Fig. 7d) between the 19 Sep. and 5 Oct. 2014. The maximum instantaneous velocity is likely to have been higher than the calculated values which are averages over 16-day periods. The surge caused a six-fold acceleration in the five months following May 2014, and more than a 20-fold acceleration since 2011/12.

After the surge peak in Sep. 2014, there was a slight deceleration which continued during winter (Fig. 7e) until maximum

5 velocities had dropped to about $1.2\,\mathrm{m\,d^{-1}}$ in April 2015. This was followed by a new phase of acceleration through May–July 2015 to almost $2\,\mathrm{m\,d^{-1}}$ in late July, slightly slower than the peak velocity in summer 2014. This acceleration came to an abrupt halt between 27 July and 7 Aug. 2015, causing the most rapid change observed with a halving of velocities over the tongue within 22 days (Fig. 8a–c). This abrupt slow-down was aligned with the lake drainage on 27 July, as indicated by the arrow in Fig. 9.

10 Deceleration continued over autumn 2015 and winter of 2015/16 and velocities almost returned to pre-surge levels with a maximum of less than $0.5\,\mathrm{m\,d^{-1}}$ in March 2016. There was a slight acceleration after April 2016 but velocities were still significantly below the previous two summers, remaining below $1\,\mathrm{m\,d^{-1}}$.

Fig. 9, consisting of a stacked time series of velocity profiles along the glacier, shows that the surge mainly affected the tongue of the glacier, between km 1 and km 8, while above the confluence (> km 8) the effect of the surge was small.



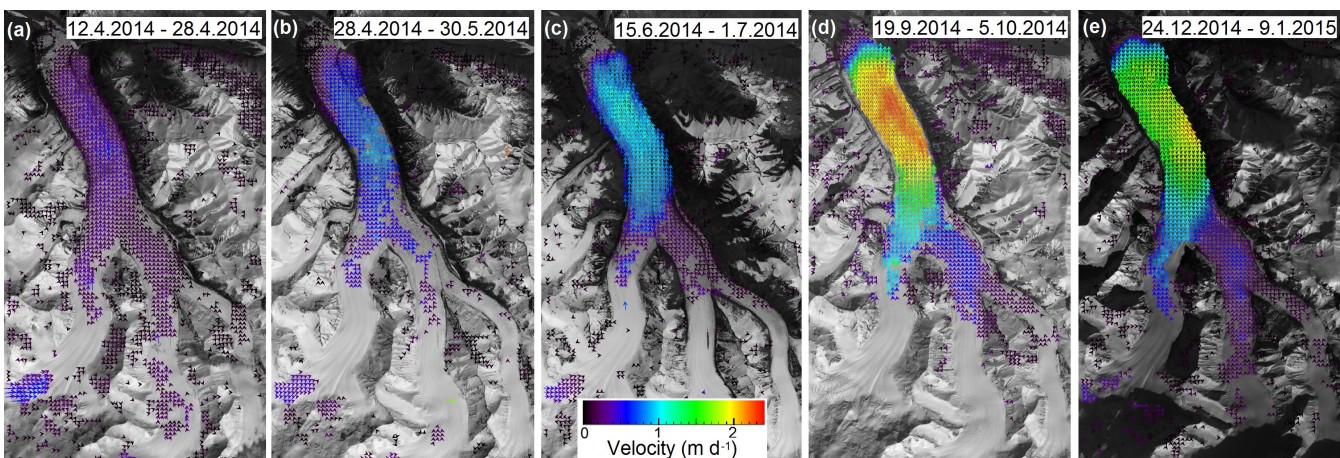

**Figure 7.** Velocity fields showing the onset and peak of the surge. Panels (a)–(c) show the initial acceleration between April and July 2014, (d) shows the maximum of the surge in Sep./Oct. followed by deceleration to lower velocities in winter 2014/15 (e). Background image from USGS Landsat 8 satellite data.

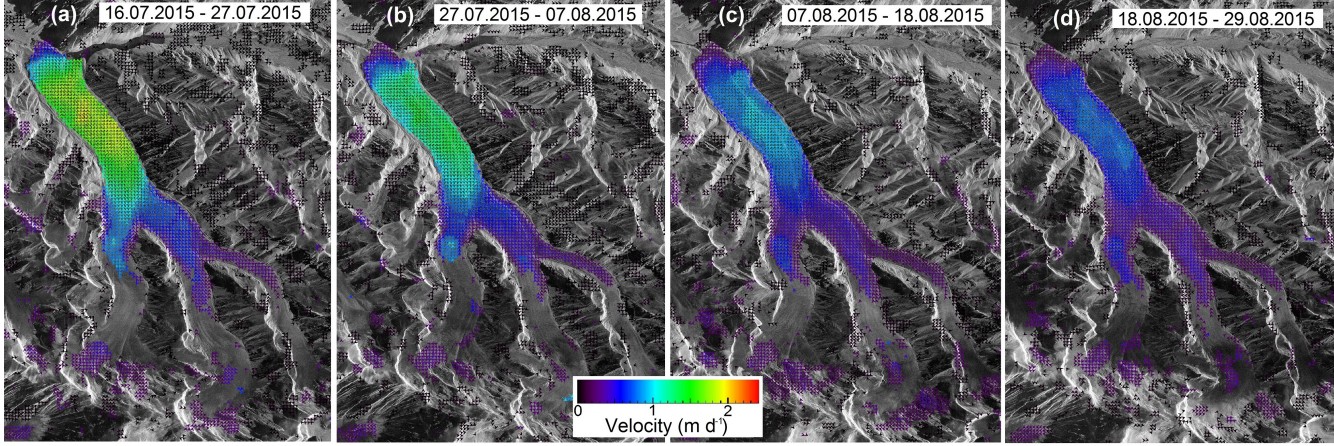

**Figure 8.** Velocity fields showing (a) the maximum velocity reached in 2015, followed by (b)–(d) the sudden deceleration from the end of July into August. These velocity fields were calculated from consecutive 11-day periods. Background image from TanDEM-X data provided by DLR.





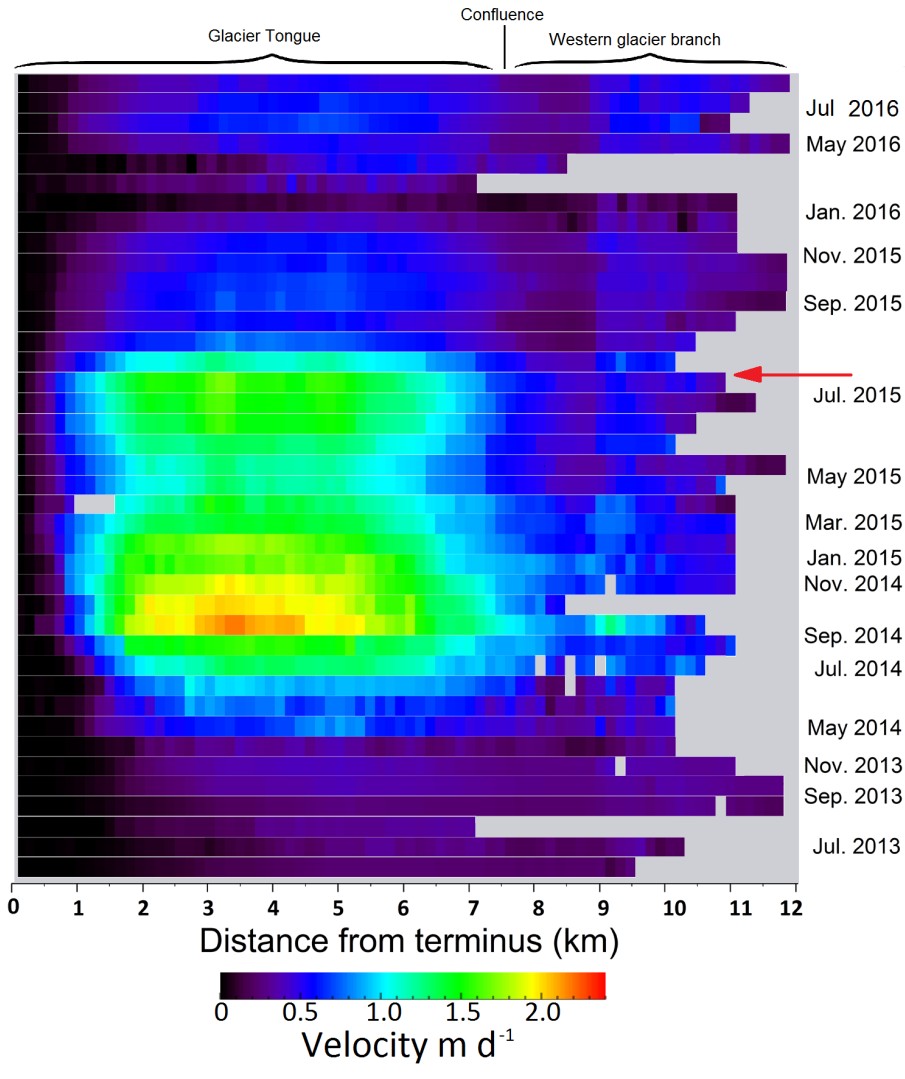

**Figure 9.** Evolution of surface velocity along the longitudinal profile (see Fig. 3) showing the spatial extent of the surge and how it evolved in time. The surge predominately affected the glacier below the confluence at km 8. The red arrow indicates the GLOF on 27 July 2015 and the corresponding abrupt deceleration.



## 4.2 Glacier surface elevation

Four DEMs based on TanDEM-X data acquired before the surge (2012–2014) and eight from after the main part of the surge (Oct.–Dec. 2015) were compared with each other and with the SRTM DEM from 2000, to reveal the dramatic changes in glacier surface elevation and, hence, ice mass distribution over Kyagar Glacier caused by the surge. Maps of elevation change over the glacier during the quiescence and surge periods are shown in Fig. 10 and 11, respectively. Longitudinal profiles of surface elevation are shown in Fig. 12a and the elevation change rates in Fig. 12b.

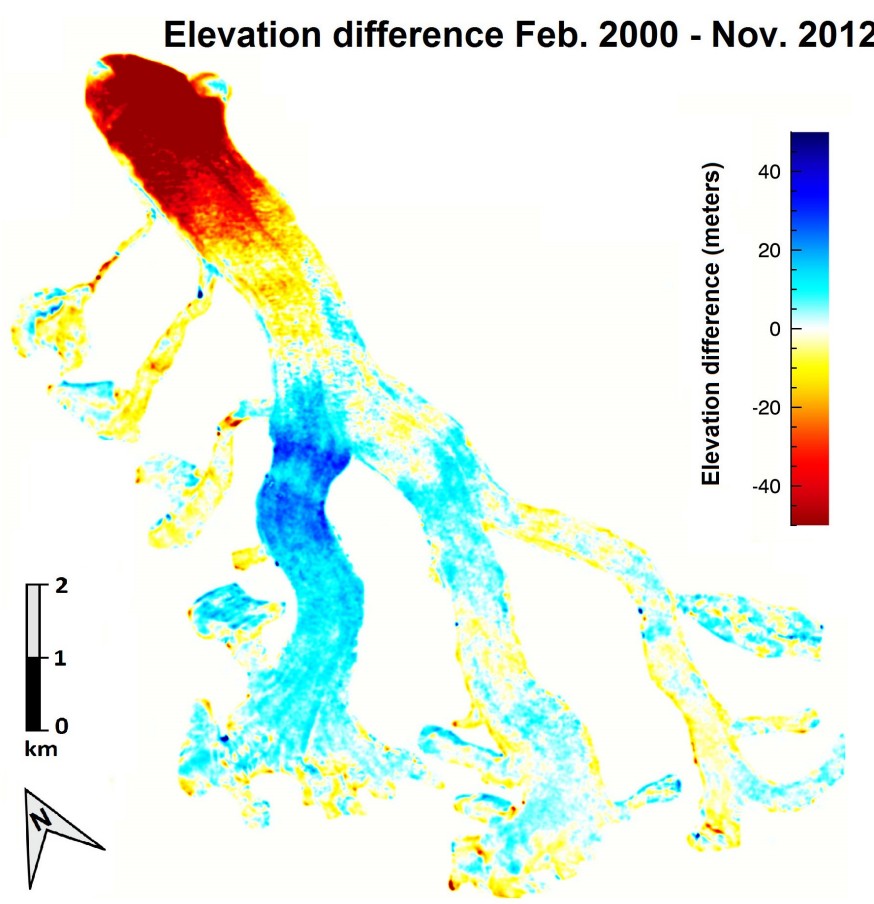

**Figure 10.** Glacier surface elevation changes from DEM subtraction between Feb. 2000 (SRTM) and Nov. 2012 (TanDEM-X). The elevation change represents 12 years of quiescence preceding the surge. The image is shown in radar coordinates for TanDEM-X data of orbit 75D, accounting for the slightly different orientation to the optical image in Fig. 3.

Before the surge, between Feb. 2000 and Nov. 2012, the surface elevation decreased gradually over most of the glacier tongue at a rate of $5 \, \mathrm{m\, a^{-1}}$ (Fig. 10 and Fig. 12b), resulting in an elevation loss of over 60 meters at the glacier terminus. At the same time, elevation increased over the western branch by up to 30 meters just above the confluence, while over the eastern



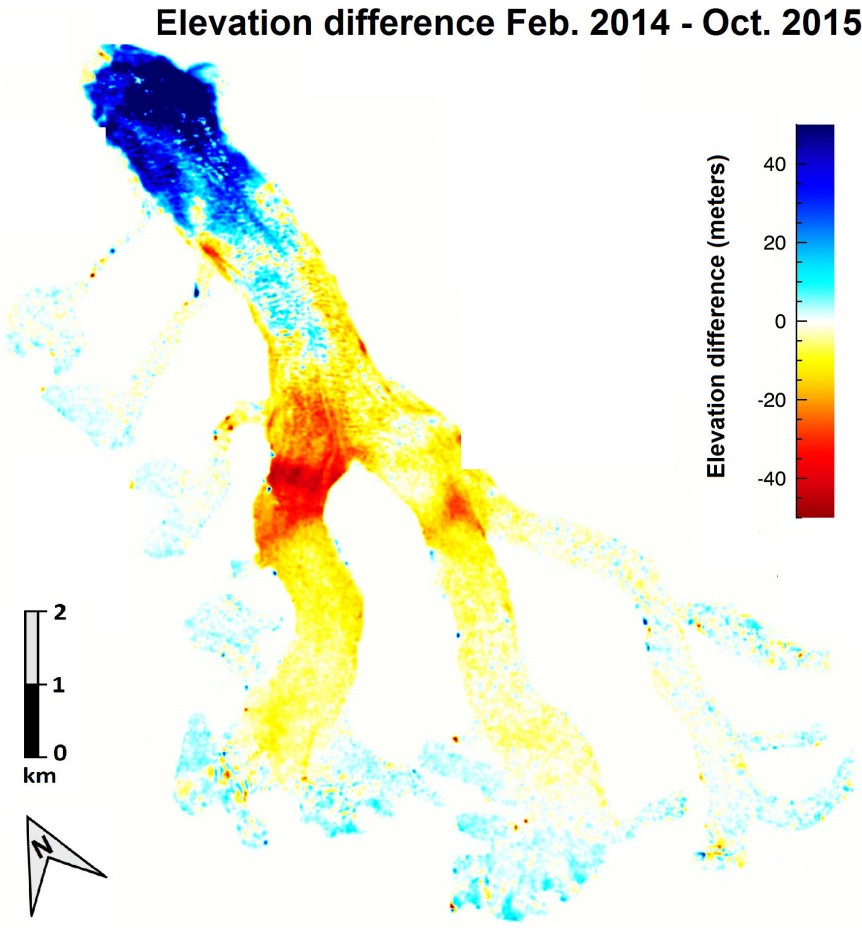

**Figure 11.** Glacier surface elevation changes during the surge from subtraction of two TanDEM-X DEMs from Feb. 2014 and Oct. 2015. This surface elevation change reverses the change pattern shown in Fig. 10 through only 1.5 years of surging.

branches the surface elevation increased more moderately with a maximum gain of 10 meters (Fig. 10). This observed pattern is typical of a surging glacier in the quiescence phase, with downwasting over the glacier tongue and ice build up in a reservoir area, which for Kyagar Glacier forms just above the confluence on the western branch.

Between 2012 and 2014, in the two years preceding the surge, there was already a slight reversal of the quiescence pattern seen in the previous 12 years, with minor elevation loss just above the confluence and mass gain over the tongue (Fig. 12), indicating mass transport down the glacier from the reservoir. During the surge in 2014/15, this mass transport from the reservoir area intensified dramatically with ice surface elevation increasing at a rate of almost $40\,\mathrm{m\,a^{-1}}$ over the lowest parts of the glacier tongue, causing thickening in excess of 80 meters at the terminus since Feb. 2014. At the same time, surface elevation decreased by more than 40 meters just above the confluence where the reservoir area formed during the quiescence. These changes are typical of a glacier surge and reflect the rapid transport of ice mass from the reservoir area down the glacier,



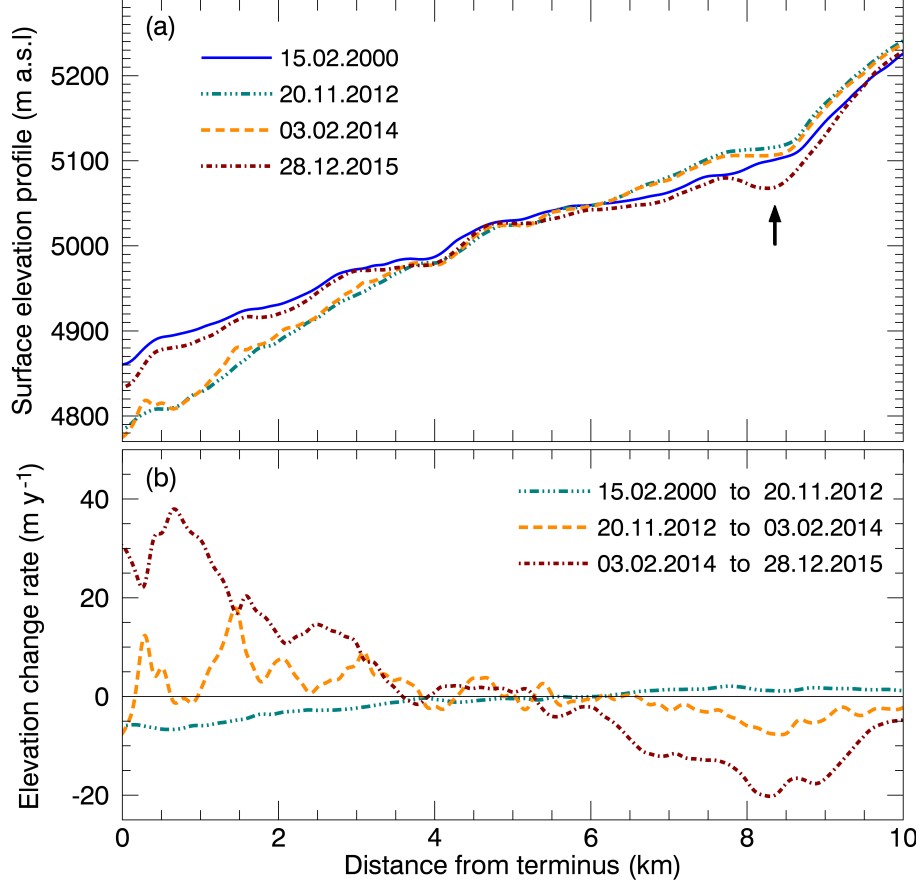

**Figure 12.** (a) Elevation profiles from 2000, 2012, 2014 and 2015, and (b) the rate of elevation change between the periods 2000–2012, 2012–2014 and 2014–2015. Profiles are taken along the transect indicated in Fig. 3 and the black arrow indicates the location of the surface hollow remaining after the surge.

in this case causing thickening at the terminus and lower glacier tongue. The mass transport during the surge essentially reversed the changes which occurred during the quiescence (2000–2012), such that the glacier surface profile at the end of 2015 had almost returned to that of 2000 (brown vs blue lines in Fig.12a).

There were significant adjustments to surface slope throughout the course of the surge, particularly over the glacier tongue. In 2000, the average slope over the first eight kilometers of the glacier was about 1.4° and by 2012/2014 it had increased to about 2.3°. By the end of 2015, after the surge, slope had decreased again to about 1.6°.

The post-surge glacier surface in Oct. 2015 showed the presence of a surface hollow, approximately 12 m deep and up to 1 km wide, at the very beginning of the western branch above the confluence just before the slope significantly steepens (Fig. 12a, indicated by arrow). Such a depression is an unusual feature but could have formed as a consequence of the surge





transporting ice from the reservoir area faster than the rate of replacement from above, owing to the observed flow disparity between the tongue and the glacier branches.

### 4.2.1 Mass balance and equilibrium line altitude

Although not directly related to the surge characterisation, we provide a geodetic mass balance estimate for Kyagar Glacier

between 2000 and 2015. The average volume difference between the SRTM DEM and eight TanDEM-X DEMs from between Oct.–Dec. 2015 was calculated and converted to mass change assuming an ice density of $850 \pm 60\,\mathrm{kg\,m^{-3}}$ (Huss, 2013). The mass balance was found to be $-0.24 \pm 0.22\,\mathrm{m\,w.e\,a^{-1}}$. For the uncertainty, the radar penetration difference between the SRTM and the TanDEM-X DEMs dominates and was estimated to be a conservative $3\,\mathrm{m}$ systematic error over the whole glacier. As the penetration for the SRTM C-band microwaves is deeper than the TanDEM-X X-band, our calculation may slightly

underestimate mass loss. On the other hand, the area used for calculation ($61\,\mathrm{km^2}$) missed some of the steepest portions of the accumulation area due to lack of coherence affecting DEM creation, possibly leading to an under-representation of the accumulation area and exaggerated mass loss. For comparison, Gardelle et al. (2013) reported an average mass balance of $+0.11 \pm 0.14\,\mathrm{m\,w.e\,a^{-1}}$ for glaciers in the east Karakoram region between 2000 and 2008.

The equilibrium line altitude (ELA) estimated from the location of the snow line at the end of the ablation period observed

from Landsat and TanDEM-X images, was $5350\pm15$, $5400\pm25$ and $5510\pm30\,\mathrm{m\,a.s.l.}$ over the western, middle and eastern branches, respectively.

### 4.3 Meteorological observations

Data from the Kyagar meteorological station (at $4800\,\mathrm{m\,a.s.l.}$) between Sep. 2012 and Jun. 2015 revealed that temperatures remained below $0°$ between October and late March to late April. Melting potential predominately occurred in June, July and

August with monthly positive degree days exceeding 150 PDDs in July and August 2013 and during the warmest months, July and August, average maximum temperatures were 4–7°C. Annual PDDs were $647°$ C, $481°$ C, $552°$ C and $528°$ C in 2013, 2014, 2015 and 2016, respectively.

### 4.4 Lake formation and drainage

Images from the monitoring station at Kyagar Glacier showed that a lake initially began forming in the river basin upstream

of the glacier terminus in early December 2014. During January and February 2015 the lake appeared to fill faster, before remaining at a constant size (still less than 1 million m³) during March. In April 2015 the lake size increased again in line with the onset of spring melting, continuing more rapidly during the summer until reaching an estimated volume of 53 million m³ before draining through subglacial channels on the 27 July 2015, as observed by TanDEM-X acquisitions (Fig. 13).

Following the drainage in July, a new lake started forming in September 2015 and remained at a volume of approximately

1.5 million m³ between Oct. and Dec. 2015. As during the previous winter the lake size increased between January and February 2016, from approximately 1.5 to 5.0 million m³, and again this winter lake filling seemed to stop during March and recom-





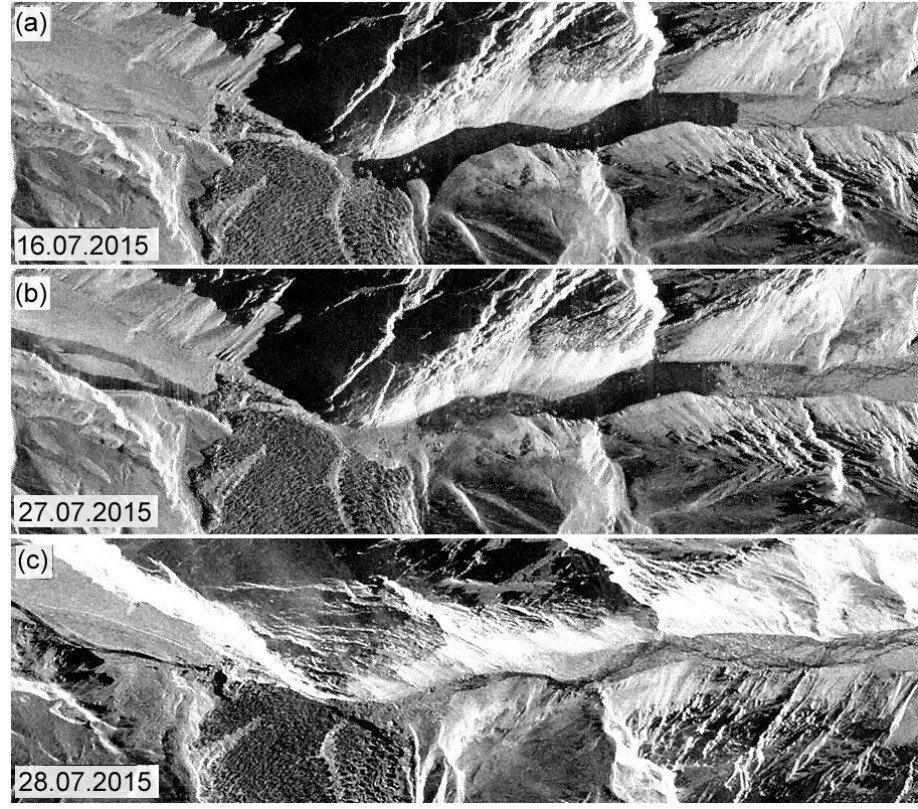

**Figure 13.** Radar backscatter images of the glacier terminus showing (a) the lake 11 days before drainage, (b) just after the start of drainage and (c) after the lake drainage. Lake drainage clearly occurred through subglacial channels, rather than through dam collapse or over topping. Images from TanDEM-X data provided by DLR.

mence with the onset of the melting season. The lake rapidly filled during summer 2016 and reached an estimated volume of 40 million m$^3$ by 13 July before a rapid drainage event occurred on 17 July 2016. Almost immediately after this event, the lake appeared to fill again and reached an estimated volume of 37 million m$^3$ on 4 Aug before a second drainage event on 11 Aug. Lake volumes, as calculated from satellite images and the lake basin DEM, are provided in the supplementary material.

## 5   Discussion

Based on the results, we discuss possible surge mechanisms producing the observed behaviour of the glacier before and during the main surge phase and rule out mechanisms which contradict the observed behaviour. The effect of the surge cycle on the GLOF hazard posed by Kyagar Glacier in the past and future is assessed to provide an outlook for its hazard potential.





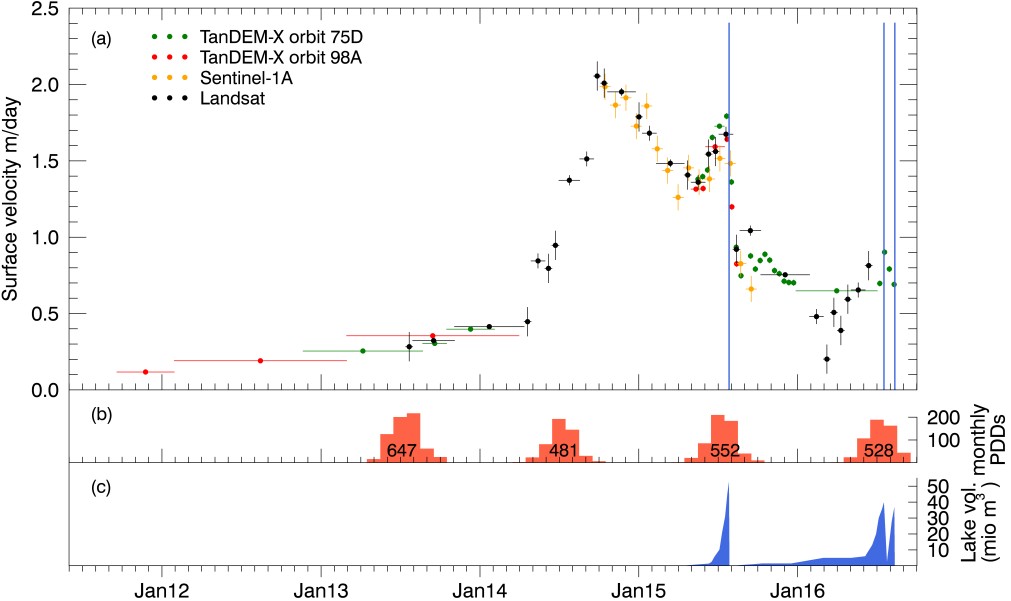

**Figure 14.** (a) Evolution of surface velocity at the middle of the glacier tongue (4 km from the terminus, Fig. 3) from the end of 2011 until mid-2016, with horizontal bars representing the period over which velocity was calculated and vertical bars showing the velocity uncertainty. The blue vertical lines indicate GLOF events. (b) Monthly PDDs indicated by the bars and yearly PDDs indicated by the numbers. (c) Temporal change in lake volume as calculated from lake extent on satellite images.

## 5.1 Surge mechanisms for Kyagar Glacier

### 5.1.1 Pre-surge build up

The observed pre-surge acceleration could have arisen through increased internal ice deformation and/or increased basal sliding, both of which may be expected following the steepening of the glacier tongue between 2000 and 2012 (Fig. 12a). The contribution of internal ice deformation $u_\mathrm{d}$ to surface flow can be estimated with the parallel-sided slab assumption with the plain strain approximation (Greve and Blatter, 2009), as

$$u_\mathrm{d} = \frac{2A}{n+1} \left(\rho g \sin \alpha\right)^n H^{n+1}, \tag{1}$$

where the strain rate factor $A = 2.4 \times 10^{-24}\,\mathrm{s}^{-1}\,\mathrm{Pa}^{-3}$ (for temperate ice, a conservative estimate), ice density $\rho = 900\,\mathrm{kg\,m}^{-3}$, Glen's exponent $n = 3$, and gravitational acceleration $g = 9.8\,\mathrm{m\,s}^{-2}$, leaving the key variables surface slope, $\alpha$, and ice thickness $H$ (Cuffey and Paterson, 2010). Assuming a constant glacier thickness of 250 m, an estimation on the high side according to the glacier bed profile presented by Haemmig et al. (2014), the 1.4° surface slope over the glacier tongue in 2000 would result in a surface velocity of 4 mm d$^{-1}$. The increased slope of 2.3° in 2012 would give deformation velocities of around 18 mm d$^{-1}$, approximately one order of magnitude lower than the observed 0.1 m d$^{-1}$ between 2011–2012 (Fig. 6). It seems that basal motion significantly contributed to flow of the glacier tongue already prior to the surge, indicating that the base of



the glacier tongue was already temperate, contradicting the thermal mechanism in which a switch from cold to temperate base causes surge onset. Conditions are different above the confluence where the surface slope of around 4.5° in 2012 could cause surface velocities on the order of $0.1\,\mathrm{m\,d^{-1}}$ through internal deformation alone, in the same order of magnitude as observed velocities. Pre-surge velocities above the confluence could therefore feasibly occur in a cold-based situation through internal

deformation without the contribution of basal motion. However, basal motion upstream of the confluence is not ruled out with this simple calculation.

The effect of a change in surface topography on basal shear stress, $\tau_b$, during the quiescence can be estimated from

$$\tau_b = \rho g H \sin \alpha, \tag{2}$$

if the glacier base is assumed to mirror the surface slope (Cuffey and Paterson, 2010). If all variables except the slope are

considered constant, the increase from 1.4° to 2.3° between 2000 and 2012 would have caused a 64% basal shear stress increase, from about 54 to 88 kPa. The thickness increase over the reservoir area would further increase $\tau_b$ at the upper part of the glacier tongue. However, the slope began decreasing between 2012 and 2014 (Fig. 12) while velocity continued to increase. This contradicts the idea that increasing slope alone could have driven the acceleration. Positive feedback mechanisms triggered by the basal stress increase must therefore play a role in the continued acceleration during the late quiescence, and

ultimately in bringing the glacier into a critical state before surge initiation. These could include increased frictional heating enhancing melt water production and water pressure at the glacier base (Dunse et al., 2015; Weertman, 1969), or increased basal deformation closing subglacial drainage channels, thus trapping water and increasing water pressure (Clarke et al., 1984; Kamb et al., 1985). Processes within the subglacial till such as a positive feedback between till deformation, consolidation and water pressure (Boulton and Zatsepin, 2006) could also play a role.

The continuation of acceleration during winter 2013/2014 (Fig. 14), rather than another slow-down as observed by Haemmig et al. (2014) during the previous winter, may indicate the presence of an inefficient subglacial drainage system. Such winter acceleration was observed prior to the 1982/83 surge of Variegated Glacier and was attributed to the establishment of an inefficient linked cavity drainage system with higher water pressure, in part due to low water flux allowing drainage channels to close (Kamb et al., 1985).

**5.1.2    Main surge phase, 2014 to 2016**

The presence of an inefficient subglacial drainage system during the last winter of the quiescence, as indicated by increased winter velocities, would account for the summer acceleration as the input of surface meltwater increased water pressure and facilitated the onset of the surge as observed in May 2014. Acceleration through the summer could reflect increasing subglacial water pressure as meltwater input continued. The deceleration after reaching maximum velocities in Oct. 2014 indicates de-

creasing subglacial water pressure, perhaps through the gradual evolution of a subglacial drainage system towards the end of summer followed by declining melt water input and possible subglacial drainage. Evidence for the drainage of en- or sub-glacially stored water during wintertime comes from the observed lake formation starting in December 2014, at a time when temperatures consistently well below zero exclude surface water sources. The lake growth and, hence, the drainage of sub-



glacial water, appeared to end in January 2015. This indicates that most subglacial water was already drained or that subglacial drainage channels closed towards the end of the winter. Closing of the subglacial channels would again put the subglacial system into a state very sensitive to surface water input and allow summer-onset acceleration in 2015. The seemingly extremely rapid response of surface velocity to the onset of surface melting indicates an efficient transfer of surface water to the glacier

base and that the glacier was in a critical state before the melt season started.

The abrupt deceleration at the end of July 2015, occurring simultaneously with the lake outburst, is a more extreme example of deceleration occurring in association with subglacial drainage. It seems that the opening of subglacial channels beneath the terminus during the lake outburst triggered the reduction of subglacial water pressure and, hence, velocity beneath the whole glacier tongue within 11 days (Fig. 8). This may have occurred through the sudden formation of an efficient drainage system

due to the change in boundary conditions at the terminus of the glacier, particularly the sudden decrease in water pressure as the lake level dropped.

### 5.1.3   Summary of surge mechanism

The various phases of the surge were facilitated by a basal motion mechanism very sensitive to subglacial water pressure, controlled by meltwater input in summer, reduced input and perhaps drainage of most of the subglacial water in early winter,

and rapid subglacial drainage during the GLOF in summer 2015. It seems that the surge is well explained by the presence of an inefficient basal drainage system facilitating high subglacial water pressure, corresponding to the mechanism proposed by Kamb et al. (1985). However, similar observations could also be explained by replacing the idea of an inefficient drainage system with a layer of subglacial till in which increased water pressure reduces till strength to a deformation threshold (Boulton and Jones, 1979; Cuffey and Paterson, 2010). It is likely that the tongue of Kyagar Glacier is underlain by a permeable till,

owing to fine-grained sedimentary rock on which the glacier tongue lies (Desio et al., 1991; Searle and Phillips, 2007). We can not speculate further on the exact nature of the subglacial drainage system as there is no field evidence, but conclude that Kyagar Glacier is a system very sensitive to water in- and outputs during the surge, rather than being purely internally regulated.

The spatial pattern of surging provides some further information about the nature of Kyagar Glacier, in particular the obser-

vation that only the tongue of the glacier participated in the surge (Fig. 9). The build up of an ice reservoir at the confluence represents the intersection between the surging tongue and the non-surging upper branches. The distinction between these two parts of the glacier is also reflected in the glacier surface slope with the tongue being much more gently-sloped (2° vs 4.5°). The surge behaviour may stem from the characteristics of the lower glacier, in particular its apparent inability to transport mass from the reservoir area down the glacier tongue to the terminus in a regular manner. Basal motion is necessary to transport

mass from the reservoir area down the glacier tongue, but some characteristic of the basal environment must cause this to occur cyclically through periods of surging. Clarke et al. (1984) noted that downstream resistance to flow may be a common factor for creating glacier surges, allowing mass build-up in a reservoir area and Björnsson et al. (2003) summarised that surge-type glaciers in Iceland tended to exhibit small slopes with velocities too slow to remain in balance with the accumulation rate. These factors could also apply to Kyagar Glacier.



We estimate a surge return period of around 15–20 years for Kyagar Glacier, based on the information that the last major period of advance was in the late 1990s (Hewitt and Liu, 2010) and backed up by the similarity between the glacier profile in 2000 and that from after the surge in 2015 (Fig. 12). The historic frequency of lake outburst flooding indicates periods of increased lake formation, and therefore probably surge activity, every 15–20 years (Fig. 2). The surge return periods of other

individual Karakoram glaciers have also been estimated at around 15 to 20 years (Mayer et al., 2011; Quincey and Luckman, 2014).

## 5.2   Future outlook for Kyagar Glacier and lake formation

The potential volume of the glacier-dammed lake at Kyagar Glacier depends both on the height of the ice dam at the glacier terminus and whether the subglacial channels through which the lake drains are open or closed. Thickening of over 60 m at

the glacier terminus caused potential GLOF volume to increase more than 40-fold since early 2014, to over 70 million m³ according to the August 2016 DEM of the glacier. GLOF hazard potential is expected to remain high for a number of years as the still slightly elevated tongue velocity continues to transport mass to the terminus area, potentially increasing the height of the ice dam until mass transport to the terminus area falls below the ablation rate. The height of the ice dam is expected to decrease at an estimated rate of $5\,\text{m}\,\text{a}^{-1}$ once mass transport to the terminus ceases, according to the ablation rate of the

latest quiescence phase and the estimated melting rate of icebergs left in the empty lake basin after lake drainage in 2009/2010 (Haemmig et al., 2014). Unless the mass balance significantly changes, it would be expected that the next quiescence phase would last until around 2030 based on an estimated 15–20 year return period.

The size of future GLOFs depends not only on the potential lake volume, but also greatly on the timing of lake drainage, as drainage may begin before the potential volume is reached. It seems that the maximum potential lake volume was not reached

in 2016 due to the outburst occurring at about 85% of the ice dam height, meaning that smaller volumes were encountered than in 2015 despite the potential volume being larger. This may be explained by subglacial channels from the 2015 lake outburst providing a weaker, preferential pathway for lake drainage and thus earlier outburst in 2016 followed by a smaller second outburst. We note that GLOFs > 80 million m³ have never been followed by a significant drainage event in the next year (Fig. 2), which perhaps indicates that large floods cause formation of subglacial channels large enough to remain open until the

following year.

It is important that the lake size and volume is monitored through satellite imagery each summer in the next few years to allow affected communities to be warned of the imminent GLOF threat.

## 6   Conclusion

Our integrative picture of the recent surge of Kyagar Glacier, built from satellite surface velocity maps, terrestrial observa-

tion station images and DEMs, provides an extraordinary insight into glacial surging in connection with surface hydrology and glacier-dammed lake formation and outburst. After gradual surface velocity increase through the last few years of the quiescence, the glacier entered a state highly sensitive to surface water input. Two dramatic acceleration phases occurred in



concurrence with the onset of the surface meltwater production in the seasons of 2014 and 2015, indicating a surge mechanism related to the evolution of the basal hydrological system and associated changes in subglacial water pressure, rather than to an internally controlled switch to warmer basal temperatures. Between the acceleration phases, deceleration was accompanied by drainage of subglacial water, evidenced by the filling of the glacier-dammed lake during the winters of 2014/15 and 2015/16.

Lake drainage in July 2015 caused instantaneous deceleration over the whole glacier tongue, indicating that a sudden drainage of the subglacial water under a large part of the glacier tongue occurred with the lake outburst event.

Surging of Kyagar Glacier is the main driver of ice-dammed lake formation and GLOFs. The thickening of over 60 m at the glacier terminus during the surge caused potential GLOF size to increase almost 40-fold since early 2014, to currently over 70 million m$^3$. The hazard potential of large GLOFs remains high in the next few years, potentially larger than the 2015

and 2016 GLOFs, but the actual magnitude depends on the timing of lake drainage. Remotely sensed data, in particular from TanDEM-X, was essential to the observation of the surge phenomenon and the assessment of hazard formation. The remote sensing of the glacier should be continued to monitor lake formation and the evolution of the ice dam height.

*Author contributions.* VR wrote major parts of the paper, calculated velocity maps and analysed all data. SL developed the offset tracking algorithms and did the interferometric processing to obtain the DEMs. MH and CH initiated the study and contributed to discussion of the

results throughout. IH provided feedback on a final version. The authors declare that they have no conflict of interest.

*Acknowledgements.* We would like to thank the German Space Agency DLR for providing TanDEM-X data through the proposal XTI_GLAC6780. We greatly appreciate the Landsat data available from the U.S. Geological Survey and the Copernicus Sentinel-1A and Sentinel-2 data from the European Space Agency ESA. Geopraevent AG provided access to data and images from the monitoring stations in China. We thank ETH Zurich for providing the funding for this study and M. Funk for comments on an earlier version.





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
