# Peer review of "Surge dynamics and lake outbursts of Kyagar Glacier, Karakoram"

_The Cryosphere, 2016_

## Referee Comment (RC1) · M. Truffer (Referee) · 29 Nov 2016

This is a very well written and interesting paper documenting most of a surge cycle of Kyagar Glacier in the Karakoram. While it is known that this mountain range has surge-type glaciers, it remains very under-studied and this paper adds a wealth of information. The satellite data coverage is amazing and allows the deduction of both elevation changes and the velocity evolution during the lead-up two a two-phase surge. The paper is essentially free of errors and well-written and could basically be published as is. I have a few small comments that should be considered for final revision:

The PDD analysis is a bit of a side-line to this paper. I do like something like it, because the availability of melt water is an important part of the story. A few more details would help: 1) It is stated that PDD is calculated from hourly data. Are the hourly data used to

calculate a daily average, or are these actually 'positive degree hours'? 2) What is the meaning of calculating PDD at one point? If the weather station is at the terminus than a day with very low positive temperatures would presumably cause melt at the very lowest part of the glacier tongue only, whereas a high degree day would cause melting over large parts of the glacier. So this measure would be a very non-linear measure of melt? 3) Does the PDD contribute more to this paper than simply a temperature graph?

Could you say a bit more of the relative role of the three tributaries to the surge? The elevation change figures indicate that perhaps all tributaries are involved in the surge? Is that also borne out in velocity evolution? In Alaska, there is distinctly different behaviors of tributaries (leading to the famous looped moraines, e.g. Clarke, 1991, J.Glac.). For a reader like me it would be interesting to know whether tributaries here play a similar role or not.

Eisen et al. (J.Glac., 2005) discuss surge initiation by a hydraulic switch that depends very sensitively on basal stress (p. 404/405). This discussion seems very relevant to this paper as well, and I recommend consulting it.

p.9, l.14/15: This is a detail, but what you're discussing is not really an error, is it? You're simply deriving the horizontal component of the velocity vector. The way you describe it you would assume that the velocity vector is surface parallel.

p.11, l.11: delete ',' (unless this involves sticking tongues into glaciers :))

p.12, l.22: The speed-up is not really uniform over the glacier tongue: the gradient gets much larger. An interesting feature is an apparent hinge point a little less than 1 km from the glacier terminus (Fig. 6). Does that correspond to something obvious on the ground?

Fig. 12: The depression in the Dec. 2015 profile is very interesting. Do you think it could be the result of a subglacial lake drainage? Sometimes these are quite recognizable in surface crevasse patterns.

p.23, l.24/25: You state that only the glacier tongue participated in the surge. This is based on the obvious velocity signature. But the elevation changes clearly show that the whole glacier is involved in the surge cycle.

Martin Truffer

---

## Referee Comment (RC2) · C. Mayer (Referee) · 6 Dec 2016

This manuscript presents an interesting combination of remote sensing and in situ observations concerning the interaction between glacier surge and glacial lake outbursts. Kyagar glacier is known to be the source of glacier lake outburst floods, but only recently it was recognized that it belongs to the family of surge type glaciers. In the upper Shaksgam Valley, glaciers reach far down the tributary valleys. These glaciers pose a potential danger by blocking the main valley and retain the river discharge in glacial dammed lakes. This combination of a surging glacier and the possibility to create a large and potentially unstable water reservoir makes Kyagar Glacier a very interesting study subject.

The authors demonstrate very well how modern remote sensing information can be

used to characterize the temporal evolution of glacier changes, not only by describing the area changes but also by inferring the dynamic situation and the mass transport. They also connect the surge termination with the characteristics of the outburst flood, which adds valuable information to the paper. In general, this paper is well structured and provides a good insight into the evolution of the surge. Even though the described surge dynamics do not reveal a new situation, this manuscript contributes very valuable information about another surging glacier in this region. There are only a few minor points I want to raise in order to hopefully improve the paper.

Specific remarks: P. 2, l. 1-3: I do not agree that the nature of glacier surging in High Mountain Asia is unknown. The mechanisms are described for different glaciers across the Karakoram and the Pamir. The recent collapse of the glaciers in Southern Tibet, just reveals that there is more to investigate about accelerating glaciers besides the known surge phenomena. P. 3, l. 4-11: This is a truly interesting relation between GLOF and surge timing. If you state that GLOFs are generally linked to the active surge phases, it might be worthwhile to mention Hoinkes (1969) who describes one of the very few other situations where the GLOF occurrence is clearly linked to surge activity: H.C. Hoinkes, 1969, Canadian Journal of Earth Sciences, 6(4), 853-861, doi:10.1139/e69-086 P. 3 l.32: It is preferably to use "North Gasherbrum Glacier" in order to distinguish from "South Gasherbrum Glacier" which flows into the Baltoro Glacier system. P.3: There should be a not in the Introduction, that the glaciers of the upper Shaksgam valley seem to be prone for surging, because apart from Kyagar and North Gasherbrum Glacier also Urdok Glacier clearly shows signs of former surge activity (e.g. Kotlyakov, 1997; Copland et al., 2011). P. 5, l.9: Is the monitoring station 600 m upstream, or 500 m as noted in the caption of Fig. 3? P. 6, l.23: What is the reason for progressively updating the master scene for the TanDEM-X data? P.7: Are you sure that the lake is only formed during surge phases? P. 9, 28/29: as the SAR system is a side-looking system, the baseline is perpendicular to the flight direction. Perpendicular to the line of sight might be misleading. P. 10, 7-11. These two sentences are somehow describing the same thing. Maybe consolidate to one sentence. P. 10/11, l.

31-35and Fig. 5: A comparison of a sequence of dry to wet images during the onset of snow melt gives an indication of penetration depth. A sequence of wet to dry conditions will not give the same results, because it is not possible to judge the snow height by remote sensing data independently. Why should a 2 m height difference between August and December indicate a 2 m penetration depth? Given that surface melt is terminated in August (no surface height change by melt and compaction afterwards), new snow on top of this surface will result in a higher surface elevation in subsequent TanDEM-X DEMs. The height difference in this case depends on the amount of snow and the snow humidity. Given that the entire snow column above the August level is dry in December, a 2 m elevation difference only indicates that there must be at least more than 2 m of snow. Unless there is a dynamic effect during this period. If the penetration depth is actually 2 m, the snow depth needs to be 4 m in order to produce a 2 m elevation change in the DEMs, which is rather unlikely for the end of December. P. 16, Fig. 10: It might be a good idea to include the longitudinal profile again in the figure and indicate the distance along the flow line. This helps to relate the velocity profiles to the elevation changes. P. 19, l. 15/16: Is there a reason for such large ELA changes over a short distance? P. 22, l. 20-24: How does this relate to the fact that the summer of 2013 probably has seen the most intensive melt amounts, according to the PDD calculations? After such an ablation season, I would expect the drainage system to be very effective. P. 23, l.4: A survey of existing photographs of Kyagar glacier back to the 1920s reveals that the surface of the glacier constantly is extremely rough and broken. This indicates that drainage of surface melt water into the glacier is rather effective. P. 23, l 24ff: this is also seen at other glaciers in the Karakoram. E.g. at North Gasherbrum Glacier also only the flat part below the ice fall is affected by the surge. P. 24, l.14: as you already have calculated the PDD sums, this relates to a realistic degree day factor of about 9 mm/$^\circ$ day. P. 24, l. 20ff: There is an interesting discussion about discharge amount and discharge seasonality in Ng et al., 2007. Climatic control on the peak discharge of glacier outburst floods, GRL, doi: 10.1029/2007GL031426

---

## Author Comment (AC1) · 22 Dec 2016

We would like to thank Martin Truffer for the insightful and positive review. It will undoubtedly help us improve a few points in the paper. We are providing this preliminary response in the spirit of interactive discussion and to allow for further feedback if needed. We address each of his points and outline how we intend to address them in the final version of the manuscript.

**The PDD analysis is a bit of a side-line to this paper. I do like something like it, because the availability of melt water is an important part of the story. A few more details would help: 1) It is stated that PDD is calculated from hourly data. Are the hourly data used to calculate a daily average, or are these actually 'positive degree hours'?**

The hourly air temperature measurements were used to create the equivalent of an average daily temperature by weighting the hourly measurements by the fraction of a day which they represent. We adjusted the text to make this clearer (p.11, l.14):

*"Positive degree days (PDD) at the glacier terminus were calculated as a proxy for potential melting. Positive air temperature measurements were summed with each measurement weighted by the fraction of a day which it represented (Vaughan, 2006), such that an hourly measurement of 6°C would contribute 0.25 PDD. The hourly air temperature data from the station at Kyagar Glacier terminus were used (...)"*

**2) What is the meaning of calculating PDD at one point? If the weather station is at the terminus than a day with very low positive temperatures would presumably cause melt at the very lowest part of the glacier tongue only, whereas a high degree day would cause melting over large parts of the glacier. So this measure would be a very non-linear measure of melt?**

Yes that's true, the calculated PDD is a value representing conditions at the terminus. The elevation range of the glacier and an estimated lapse rate can be used to make a short statement about the expected melting period over the majority of the glacier rather than just at the terminus. We will rephrase the PDD section in 4.3 Meteorological observations on p.19 to clarify the points raised:

*"Temperatures remained below 0°C between mid-October and late April according to data from the meteorological station at the glacier terminus (at 4800 m a.s.l.). The warmest months, July and August, experienced average daily maximum temperatures of 4–7°C and monthly PDDs exceeding 150 at the glacier terminus. By taking into account the elevation of the glacier surface and an approximate lapse rate of 0.006°C m$^{-1}$, it can be inferred that over the whole glacier tongue, PDDs are positive between May and October, whilst over the bulk of the accumulation area (about 900 m above the terminus) melt potential was only significant from June to August. Evidence of high-altitude melt is seen in the TanDEM-X backscatter images from August 2015 (Figure*

*5). Annual PDDs at the glacier terminus were 647°C, 481°C, 552°C and 528°C in 2013, 2014, 2015 and 2016, respectively."*

**3) Does the PDD contribute more to this paper than simply a temperature graph?**

We include the PDD analysis primarily as a way to present the temperature data. We believe it gives a better overview of the temporal distribution of melt potential than a temperature plot, and it provides annual or monthly values which can easily be compared.

**Eisen et al. (J.Glac., 2005) discuss surge initiation by a hydraulic switch that depends very sensitively on basal stress (p. 404/405). This discussion seems very relevant to this paper as well, and I recommend consulting it.**

Thanks for this suggestion. The discussion on the sensitivity of the drainage system to increased basal stress is a very relevant reference which we shall add to p.22, l.11:

*"Given the potential sensitivity of the subglacial drainage efficiency to basal stress (Eisen et al., 2005) the conditions at the end of the quiescence phase could be expected to favour the switch to an inefficient drainage system."*

Also after reading Eisen et al. (2005) we intend to add this additional text to the discussion p.23, l.17, as we find the difference in seasonality of initiation and termination between many Alaskan surges and Kyagar Glacier worth noting:

*"It seems that the surge is well explained by the presence of an inefficient basal drainage system facilitating high subglacial water pressure, corresponding to the mechanism proposed by Kamb et al. (1985). However, the seasonality observed at Kyagar Glacier is different to the often cited winter initiation associated with closure of subglacial channels in the hydrological switch mechanism (Eisen et al. 2005, Kamb et al., 1985). In the case of Kyagar Glacier, development of an inefficient drainage system in winter does not necessarily facilitate increased subglacial water pressure until the beginning of the melt season, due a lack of liquid water in winter. Surge initiation in winter*

*should not be considered a precondition of hydrologically controlled surging (see e.g., Jiskoot & Low, 2011)."*

**p.9, l.14/15: This is a detail, but what you're discussing is not really an error, is it? You're simply deriving the horizontal component of the velocity vector. The way you describe it you would assume that the velocity vector is surface parallel.**

True, it makes more sense to use the term 'velocity difference' rather than 'velocity error'. This will be corrected in the manuscript.

**p.11, l.11: delete ',' (unless this involves sticking tongues into glaciers :) )**

The offending comma will be removed to avoid misunderstanding!

**p.12, l.22: The speed-up is not really uniform over the glacier tongue: the gradient gets much larger. An interesting feature is an apparent hinge point a little less than 1 km from the glacier terminus (Fig. 6). Does that correspond to something obvious on the ground?**

What we mean to say here is that the spatial pattern of acceleration is rather uniform over the tongue – there is no 'surge front' travelling down-glacier, as has been observed in some other Karakoram Glaciers (e.g. Quincey 2011, 2015). We shall word it slightly differently, p.12, l.19–24:

*" In the 2.5 years before surge onset, a gradual but clear acceleration occurred, greatest over the middle of the glacier tongue (between km 3 and km 6) with an increase in velocity from 0.1 m d$^{-1}$ in winter 2011/12 to over 0.4 m d$^{-1}$ in winter 2013/14 (Fig. 6). The location of the maximum velocity moved from above the confluence at km 10 at the end of 2011 to over the glacier tongue at km 5 in 2013/2014. Apart from this early shift, the spatial pattern of acceleration over the glacier tongue was quite uniform with no evidence of a 'surge front' of acceleration moving down the glacier, as observed for some other Karakoram glaciers (Mayer et al., 2011; Quincey et al. 2015)."*

The almost negligible acceleration over the lowest 1km of the glacier (resulting in this

apparent 'hinge point' above which acceleration becomes evident) arises because horizontal flow is impeded by the mountain flank against which the glacier terminus pushes. This is why we observe so much thickening at the terminus, rather than horizontal advance. There is a conversion to vertical velocity which is not visible in our horizontal velocity assessment.

**Fig. 12: The depression in the Dec. 2015 profile is very interesting. Do you think it could be the result of a subglacial lake drainage? Sometimes these are quite recognizable in surface crevasse patterns.**

The suggestion that the surface depression could have come about through the drainage of a subglacial lake is an interesting one. However, after looking again closely at crevasse patterns over the area, we don't see any evidence of subglacial lake drainage. There are very distinctive transverse crevasses across the steep slope immediately up from where the depression formed after the surge. However, these are rather indicative of extensional stress in the flow direction, and their enlargement after the surge is likely a result of the rapid steepening over this part of the glacier as mass is removed from the reservoir area at the bottom of this steep slope (see Fig. 1 below).

We think that the depression forms because of the divergence in speed between the glacier tongue below and the tributary above, causing 'emptying' of the reservoir area during the surge. A very similar formation was observed at the Belvedere Glacier in Italy, after a surge of the glacier tongue away from the steeper upper slopes in 2000/2002 (Haeberli et al., 2002; Kääb et al., 2004, page I/70)

**Could you say a bit more of the relative role of the three tributaries to the surge? The elevation change figures indicate that perhaps all tributaries are involved in the surge? Is that also borne out in velocity evolution? In Alaska, there is distinctly different behaviors of tributaries (leading to the famous looped moraines, e.g. Clarke, 1991, J.Glac.). For a reader like me it would be interesting to know whether tributaries here play a similar role or not.**
Looped moraines are expected when tributaries surge into a non-surging part of the glacier, or two joining branches surge at different times. In the case of Kyagar glacier it seems to be the glacier tongue below the confluence which is surging away from the tributaries and we do not see any evidence that the tributaries themselves surge independently. The attached video (supplement to this comment) very nicely shows the lack of looped moraine formation. We will add a short explanation for the absence of looped moraines to make this clear in our discussion (see text added in response to the next comment).

**p.23, l.24/25: You state that only the glacier tongue participated in the surge. This is based on the obvious velocity signature. But the elevation changes clearly show that the whole glacier is involved in the surge cycle.**

The greatest acceleration was observed over the glacier tongue, but the main evidence which leads us to believe that the surge behaviour predominately occurs over the glacier tongue was the pattern of mass distribution change from the DEMs. The instability which develops during quiescence is caused in part by the buildup of mass at the top of the surging area, which in the case of Kyagar Glacier is at the bottom of the tributaries/top of the glacier tongue. It is the glacier tongue which develops the characteristic steepening during quiescence and massive redistribution during the surge. On the other hand, the tributaries don't show such irregular behavior. They experience slight thickening all over during quiescence (and large thickening at the base of the western tributary, the main reservoir area), and slight thinning over the surge (again, large thinning only over the reservoir area). We consider the slight acceleration and thinning of the tributaries during the surge as a 'side effect' of the glacier tongue surge. Although we don't consider the tributaries as primarily surging, they are involved through providing mass to the reservoir areas and are affected by the surging tongue. We will amend the text at p.23, l.24/25 to reflect this:

*"The spatial pattern of acceleration and elevation change over Kyagar Glacier provides further information about the nature of the surge, in particular that it was the tongue*

TCD

Interactive
comment

*of the glacier which primarily underwent surging, evidenced by the velocity increase (Fig. 9) and the steepening of the profile over the glacier tongue during quiescence (Fig. 12). The build-up of an ice reservoir at the confluence represents the intersection between the tongue, which develops an unstable profile during the quiescence followed by dramatic surging, and the tributaries, which maintain more steady flow and support the recharge of the ice reservoir during the quiescence. We note also that looped moraines do not form at Kyagar Glacier because the tributaries are not surging into a non-surging part of the glacier. The actual surge activity only seems to affect the glacier tongue below the confluence."*

REFERENCES:

Kääb, A., Huggel, C., Barbero, S., Chiarle, M., Cordola, M., Epifani, F., Haeberli, W., Mortara, G., Semino, G., Semino, P., Tamburini, A. and Viazzo, G.: *Glacier hazards at Belvedere Glacier and the Monte Rosa East Face, Italian Alps: Processes and mitigation*, International Symposon Intrapraevent 2004 – Riva/Trient, Available: http://folk.uio.no/kaeaeb/publications/inter04.pdf, 2004

Haeberli, W., Kääb, A., Paul, F., Chiarle, M., Mortara, G., Mazza, A., Deline, P. and Richardson, S.: *A surge-type movement at Ghiacciaio del Belvedere and a developing slope instability in the east face of Monte Rosa, Macugnaga, Italian Alps*, Norwegian Journal of Geography, 56(2), 104-111, doi:10.1080/002919502760056422, 2002

Please also note the supplement to this comment:
http://www.the-cryosphere-discuss.net/tc-2016-236/tc-2016-236-AC1-supplement.zip
* * *
[Figure]

[Figure]

**Fig. 1.** TanDEM-X images showing development of transverse crevasses and circling the approximate location of the surface depression which formed at around km 8.5.

---

## Author Comment (AC2) · 4 Jan 2017

We would like to thank Christoph Mayer for his review. It is particularly useful to have this input given his familiarity with the surge of the nearby North Gasherbrum Glacier. In this reply we provide responses to each of his points and indicate the changes we intend to make to the manuscript.

**There are only a few minor points I want to raise in order to hopefully improve the paper. Specific remarks:**

**P. 2, l. 1-3: I do not agree that the nature of glacier surging in High Mountain Asia is unknown. The mechanisms are described for different glaciers across the Karakoram and the Pamir. The recent collapse of the glaciers in Southern Tibet, just reveals that there is more to investigate about accelerating glaciers**

**besides the known surge phenomena.**

Perhaps the statement was a bit of an exaggeration as there have been a number of detailed studies on surging in the region, so we have adjusted the text accordingly. By including the reference to the Tibet glacier collapse example we want to point out that there can still be 'surprises' related to glacier instabilities in the region.

*"While surging glaciers in North America and Svalbard have been investigated in considerable detail, the large concentration of surge-type glaciers existing in the central Asian mountains, including the Karakoram (Copland et al, 2011) are relatively under studied. Improved understanding of surge glacier dynamics in this region can assist anticipation of glacier behaviour and hazard development in the future. The recent unprecedented collapse of two surging glaciers in Tibet (GAPHAZ, 2016) highlights the potentially unexpected nature of glacier instabilities in the region."*

**P. 3, l. 4-11: This is a truly interesting relation between GLOF and surge timing. If you state that GLOFs are generally linked to the active surge phases, it might be worthwhile to mention Hoinkes (1969) who describes one of the very few other situations where the GLOF occurrence is clearly linked to surge activity: H.C. Hoinkes, 1969, Canadian Journal of Earth Sciences, 6(4), 853-861, doi:10.1139/e69-086**

Thank you for the suggestion, this was an interesting read and we shall include a brief reference to it at p.3, l.11:

*"Recurring GLOFs linked to periods of glacier surging have also been observed for other surging glaciers (e.g. Hoinkes, 1969)."*

**P. 3 l.32: It is preferably to use "North Gasherbrum Glacier" in order to distinguish from "South Gasherbrum Glacier" which flows into the Baltoro Glacier system.**

Thank you for the clarification, we will use the name North Gasherbrum Glacier.

**P.3: There should be a note in the Introduction that the glaciers of the upper Shaksgam valley seem to be prone for surging, because apart from Kyagar and North Gasherbrum Glacier also Urdok Glacier clearly shows signs of former surge activity (e.g. Kotlyakov, 1997; Copland et al., 2011).**

This is a good point which we will include in the discussion (rather than the introduction) p.23, l.34:

*"The fact that at least three of the five closest downstream neighbouring glaciers also experienced surging (Copland et al. 2011; Mayer et al. 2011; Quincey et al. 2015) also indicates possible locational influences on surging, for example due to local climatic characteristics (Sevestre and Benn, 2015)."*

**P. 5, l.9: Is the monitoring station 600 m upstream, or 500 m as noted in the caption of Fig. 3?**

This will be changed to "about 500m upstream" in both instances. 500m is the best estimate of the distance of the station from the upstream ice margin of the terminus, although it is an approximate value because the position of the terminus is not constant.

**P. 6, l.23: What is the reason for progressively updating the master scene for the TanDEM-X data?**

A different co-registration algorithm was used for TanDEM-X data, one which updates the master scene as an average of all previously co-registered scenes, to reduce speckle and temporally average snowmelt and glacier movement. We shall mention this:

*"(. . .) for TanDEM-X, another co-registration algorithm was used where the master was updated progressively as the average of all previously co-registered slave scenes in order to temporally smooth out moving features (e.g. crevasses on glaciers) or strongly changing patterns (e.g. snowmelt)."*

**P.7: Are you sure that the lake is only formed during surge phases?**

Not at all. In fact we are quite sure that a lake can also form when the glacier is not actively surging (e.g. Fig. 2, 2009), but surging causes larger potential lake size and more probable lake formation. Periods of larger and more frequent outbursts seem to coincide with periods of suspected surging (p.3, l.7). We don't intend to give the impression that the lake exclusively forms during surge phase. Our rephrasing of the caption of Fig. 4 should avoid giving this impression:

*"Images from the observation station upstream of Kyagar Glacier's terminus from (a) before and (b) during the surge. The glacier, flowing from left to right, blocks the flow of the river and causes lake formation. The dashed line in (b) indicates the ice dam height from 2012 (a), highlighting the dramatic thickening at the terminus."*

**P. 9, 28/29: as the SAR system is a side-looking system, the baseline is perpendicular to the flight direction. Perpendicular to the line of sight might be misleading.**

The total baseline, i.e. the distance between both satellites, can be composed into three perpendicular components: along track ($B_{||}$), parallel to line of sight (range offset), and perpendicular to the line of sight $B_{\perp}$). The two latter components form the across-track separation. We propose this slight reformulation:

*"The phase gradient and hence the DEM accuracy depends on the perpendicular interferometric baseline $B_{\perp}$, which is the component of the distance between the two SAR satellites which is perpendicular to both the line-of-sight and the flight direction."*

**P. 10, 7-11. These two sentences are somehow describing the same thing. Maybe consolidate to one sentence.**

There was some repetition which has been removed:

*"The extremely rough glacier surface topography, with ice pinnacles up to 40 m high and 20-4 0m apart (estimated from shadow lengths and the observations from Haemmig et al. (2014)), caused strong decorrelation and phase wraps within the coherence*

*window of 15x15 m, meaning that DEMs could not be created over the glacier tongue with baselines $B_\perp$ < 200 m (HoAs below 20m)."*

**P. 10/11, l.31-35 and Fig. 5: A comparison of a sequence of dry to wet images during the onset of snow melt gives an indication of penetration depth.**

That is correct. This was done to estimate the penetration depth at the onset of snowmelt 2015 as commented in the paper p.10, l.31-33.

**A sequence of wet to dry conditions will not give the same results, because it is not possible to judge the snow height by remote sensing data independently. Why should a 2 m height difference between August and December indicate a 2 m penetration depth? Given that surface melt is terminated in August (no surface height change by melt and compaction afterwards), new snow on top of this surface will result in a higher surface elevation in subsequent TanDEM-X DEMs. The height difference in this case depends on the amount of snow and the snow humidity. Given that the entire snow column above the August level is dry in December, a 2 m elevation difference only indicates that there must be at least more than 2 m of snow.**

We totally agree for the case when the elevation change is positive. We forgot to mention, that we observed a negative elevation change (see answer below).

**Unless there is a dynamic effect during this period. If the penetration depth is actually 2 m, the snow depth needs to be 4 m in order to produce a 2 m elevation change in the DEMs, which is rather unlikely for the end of December.**

An apparent 2 m height decrease between small baseline DEMs from Aug 2015 (Fig. 5a, wet snow, low backscatter) and December 2015 (Fig. 5b, dry snow, high backscatter) indicated a penetration depth of approximately 2 m into the refrozen summer snow ONLY IF elevation decrease from subsidence and compaction and elevation increase from snowfall are neglected for the four month period between the images. We have

come to the conclusion that this is an unreasonable assumption and that it's true that we can't infer much about penetration depth from the image pair with such large temporal separation. We decided to only use the image pair with short temporal separation at the onset of snowmelt.

We therefore reformulate the whole paragraph from p.10 ,l.28 – p.11,l.3 as follows:

*"Over the tongue of Kyagar Glacier, the backscatter intensity changed little between seasons (<5 dB), because infrequent snowfall means that the bare ice surface roughness dominates the backscatter signal from the tongue. Penetration is therefore expected to be negligible over the glacier tongue. In contrast, large seasonal changes in backscatter intensity indicate changing water content and thus varying penetration depths over the accumulation basin. Backscatter decreased by more than 10 dB at the onset of snowmelt in 2015 over the accumulation areas, and an apparent surface height increase of less than 2 m was calculated between two large baseline interferograms from before snowmelt (2015-06-02) and at the onset of snowmelt (2015-06-13). This indicates a TanDEM-X penetration depth of 2 m or less in dry snow conditions over the upper glacier. The relatively small penetration depths in the accumulation area might be a result of ice lenses formed by refreezing after strong melt events extending to over 6000 m a.s.l. in August, a phenomenon also observed by Dehecq at al. (2015). Figures showing the backscatter intensity changes are included in the supplementary material."*

**P. 16, Fig. 10: It might be a good idea to include the longitudinal profile again in the figure and indicate the distance along the flow line. This helps to relate the velocity profiles to the elevation changes.**

We will add this to Figures 10 and 11.

**P. 19, l. 15/16: Is there a reason for such large ELA changes over a short distance?**

A possible reason for some of the difference in ELA between the three glacier branches could be differences in snow redistribution, wind drift, or precipitation, as temperature and radiation are likely to be very similar for all branches. However, the manual estimation of ELA from optical and radar images is rather subjective and the margins of error which we supplied in the manuscript were too low to reflect this uncertainty. After looking again at the ELA estimations is seems that an error margin of ±80m is more appropriate. We rephrase p.19, l.14-16:

*"The equilibrium line altitude (ELA) estimated from the location of the snow line at the end of the ablation period observed from Landsat and TanDEM-X images, was 5350 ±80, 5400 ±80 and 5510 ±80 m a.s.l. over the western, middle and eastern branches, respectively."*

**P. 22, l. 20-24: How does this relate to the fact that the summer of 2013 probably has seen the most intensive melt amounts, according to the PDD calculations? After such an ablation season, I would expect the drainage system to be very effective.**

2013 did have higher melt potential in summer and autumn than the three years which followed, so one would indeed expect that an efficient drainage system would be more likely to form in 2013. Perhaps the gradual increase in basal sliding which took place over the two or so years before the surge hindered the formation of an efficient enough drainage system, despite possibly larger meltwater input in the summer before the surge started.

**P. 23, l.4: A survey of existing photographs of Kyagar glacier back to the 1920s reveals that the surface of the glacier constantly is extremely rough and broken. This indicates that drainage of surface melt water into the glacier is rather effective.**

Yes, the deeply pinnacled, crevassed surface is very important feature of the glacier and we also think this may assist vertical drainage. We have however noticed on

the high-resolution Sentinel-2 optical images from 2016 (one of which we will add as a supplementary figure and is included at the end of this document), that some meltwater ponds form on the glacier surface between the pinnacles during summer, indicating that some meltwater at least is not well connected to vertical drainage channels (and does not percolate through the cold ice). However, we still think that there must be sufficient vertical drainage cracks/channels to allow meltwater to reach the glacier base. We will insert the following text on p.23, l.3:

*"The seemingly extremely rapid response of surface velocity to the onset of surface melting indicates an efficient transfer of surface water to the glacier base which was in a critical state before the melt season started. The heavily crevassed surface, as observed during past expeditions (Mason 1928; Haemmig et al., 2014) and seen on remotely sensed images, may significantly contribute to the efficiency of vertical drainage. We note, however, that on some images supraglacial lakes are present on the glacier surface (Fig. 3 in supplementary material). This observation might indicate that surface water is not always connected with the subglacial drainage system despite of extensive crevassing. Based on the available evidence, we can however also not rule out the possibility that the supraglacial lakes are an expression of high englacial water pressures during the surge."*

**P. 23, l 24ff: this is also seen at other glaciers in the Karakoram. E.g. at North Gasherbrum Glacier also only the flat part below the ice fall is affected by the surge.**

We add the following sentence at p.23, l.25:

*"Surging confined mainly to the flatter, lower part of the glacier has been observed for a number other Karakoram surges (Mayer et al, 2011; Quincey et al., 2015)."*

**P. 24, l.14: as you already have calculated the PDD sums, this relates to a realistic degree day factor of about 9 mm/day.**

This is a nice comparison and a confirmation of the melt estimate, thanks for bringing it to our attention. We will briefly mention it when we present the PDD results, p.19, l.22:

*"The melt rate at the terminus is estimated to be around 5 m a$^{-1}$, according to the terminus surface elevation decrease during quiescence (Fig. 12) and the melt rate of icebergs left in the empty lake basin after lake drainage in 2009 (Haemmig et al. 2014). Combining this melt rate and an average of 552 PDDs annually gives a realistic degree-day factor of about 9 mm w.e. $^{\circ}$C$^{-1}$ d$^{-1}$."*

**P. 24, l. 20ff: There is an interesting discussion about discharge amount and discharge seasonality in Ng et al., 2007. Climatic control on the peak discharge of glacier outburst floods, GRL, doi: 10.1029/2007GL031426**

Thanks for pointing out another relevant reference. Ng et al. (2007) show that higher temperatures during GLOF events cause higher peak discharges through increasing meltwater supply rate and lake water temperature. This effect may also impact the peak discharge during GLOFs from Kyagar Glacier, but we think the most significant factor for GLOFs from Kyagar Glacier are the glacier surge dynamics and the properties of the ice dam (height, existence of drainage channels beneath the terminus etc.). We plan to mention this reference briefly as follows at p.24, l.25:

*"Meteorological factors such as air temperature during the GLOF may also influence the peak flood discharge (Ng et al. 2007)."*
* * *
[Figure]

**Fig. 1.** Optical Sentinel-2 image from 27.05.2016 showing the presence of supraglacial lakes.

---

## Author Response (AR2)

**Author response to Editor comments on *Surge Dynamics and lake outbursts of Kyagar Glacier, Karakoram**

Dear Olaf Eisen,

Thank you for assessing and accepting our paper, and for the further suggestions. Here is our response to your comments, followed by the marked up latexdiff showing the changes made to the manuscript.

**HoA vs HA: both are used, inconsistent. Please clarify.**

We corrected the abbreviation for height of ambiguity to HoA (p.6, l.28).

**The statement in second last paragraph of 5.1.3, p13 l85ff "indicates possible locational influences on surging, for example due to local topographic and climatic conditions (Sevestre and Benn, 2015)." reads somewhat unspecific. Of course local topography influences a glacier. I assume you would like to stress that certain characteristics of glaciers in this mountain range are comparable, such as topography distribution, climate and likely also geology, and that such regional commonalities might be important that several of these glaciers surge. Please rephrase and clarify accordingly.**

Thanks for bringing this up. Yes, we mean the topography distribution (steep accumulation area, relatively flat tongues), underlying sedimentary geology and possibly characteristics of the common climate, and have clarified this a bit better in the current manuscript as follows: *At least three of the five closest downstream neighbouring glaciers have also experienced surging (Mayer et al., 2011; Quincey and Luckman, 2014), and share common topographic and climatic factors possibly contributing to their surge nature. The combination of steep accumulation areas and relatively flat glacier tongues may be conducive to surge development, as may the underlying sedimentary geology or the particular climatic conditions according to the climatic envelope theory of Sevestre and Benn (2015).*

**Acknowledgements: The two referees each provided a thorough review and pointed out several issues published in the literature you were not aware of, but which are important to the overall scope of this paper. I suggest to acknowledge this here.**

Thanks for pointing out this important omission, we have added a note thanking the reviewers as their reviews were very helpful and improved the depth of the paper.

In addition to addressing your comments, we have made a couple of other changes as follows:

1) The velocity time series in Figure 13 has been updated to include six additional velocities from September to December 2016. The supplementary list of acquisitions used was also updated to reflect the additional data, and mention of the temporal range of velocity data is corrected in the manuscript (p.4 l.8, p.7 l.34, p.8 l.12 in diff file).

2) We have made a new observation about a supraglacial lake on the eastern edge of the glacier, near the confluence, which is well visible with TanDEM-X backscatter images. This lake becomes visible about a month before the GLOF in 2015, was present during the GLOF but had disappeared by the day after. In 2016 this lake again appeared, a bit over a month before the first 2016 GLOF and disappeared after the outburst. We see the presence of this small lake as an expression of high englacial water pressure before the GLOFs, at which time glacier velocity was heightened. The disappearance of the small lake after the GLOFs indicates the establishment of more efficient en/subglacial drainage and corresponds with the deceleration of the glacier following the GLOF. We have added a paragraph about this observation in the lake drainage results section 4.4, p.11, a mention in the discussion section 5.1.2 p.13, and Figure 1 below is to be included in the supplementary figures. We hope that you agree this is a valuable addition to the paper, even if it comes rather late in the submission process.

[Figure]

Figure 1: This image is to go in the supplementary figures with the following caption: A series of TanDEM-X backscatter intensity images showing a small lake on the eastern edge of the glacier (indicated by the inset box on the full image) which was present before and absent after the GLOF events in 2015 and 2016. The presence of the supraglacial lake is an expression of high englacial water pressure during the surge and its disappearance after the GLOFs strongly indicates the establishment of more efficient en/subglacial drainage.

Many thanks again, on behalf of the authors,

Vanessa Round

[revised manuscript text omitted]